# A New Basal Neornithischian Dinosaur from the Phu Kradung Formation (Upper Jurassic) of Northeastern Thailand

Sita Manitkoon [1,*], Uthumporn Deesri [1,2], Bouziane Khalloufi [1], Thanit Nonsrirach [1], Varavudh Suteethorn [3], Phornphen Chanthasit [4], Wansiri Boonla [4] and Eric Buffetaut [5]

1 Palaeontological Research and Education Centre, Mahasarakham University, Khamrieng, Maha Sarakham 44150, Thailand
2 Faculty of Science, Mahasarakham University, Khamrieng, Maha Sarakham 44150, Thailand
3 Dinosaur Research Unit, Mahasarakham University, Maha Sarakham 44150, Thailand
4 Sirindhorn Museum Department of Mineral Resources, Sahatsakhan, Kalasin 46140, Thailand
5 Laboratoire de Géologie de l'Ecole Normale Supérieure, CNRS (UMR 8538), Paris Sciences et Lettres Research University, 24 rue Lhomond, 75231 Paris Cedex 05, France
* Correspondence: sita.m@msu.ac.th or sita_aqua@hotmail.com

**Abstract:** An exceptional articulated skeleton of a new basal neornithischian dinosaur, *Minimocursor phunoiensis* gen. et sp. nov., was discovered in the Late Jurassic Phu Kradung Formation at the Phu Noi locality, Kalasin Province, Thailand, a highly productive non-marine fossil vertebrate locality of the Khorat Plateau. It is one of the best-preserved dinosaurs ever found in Southeast Asia. *Minimocursor phunoiensis* gen. et sp. nov. shows a combination of both plesiomorphic and apomorphic characters resembling those of Late Jurassic to Early Cretaceous small-bodied ornithischians from China: a low subtriangular boss is projected laterally on the surface of the jugal, the brevis shelf of the ilium is visible in lateral view along its entire length, a distinct supraacetabular flange is present on the pubic peduncle of the ilium, the prepubis tip extends beyond the distal end of the preacetabular process of the ilium, and the manus digit formula is ?-3-4-3-2. The phylogenetic analysis shows that this dinosaur is among the most basal neornithischians. This study provides a better understanding of the early evolution and taxonomic diversity of ornithischians in Southeast Asia.

**Keywords:** ornithischia; Southeast Asia; Khorat Plateau; Late Jurassic; *Minimocursor phunoiensis*

## 1. Introduction

### 1.1. The Neornithischia

Neornithischia is a clade of herbivorous dinosaurs including ornithopods, marginocephalians and a diversity of small bipedal basal forms that were historically referred to as hypsilophodontids [1–3]. The classification of basal neornithischians was presented and debated by many palaeontologists [4–8]. In Asia, the earliest neornithischian is *Sanxiasaurus modaoxiensis* from the Middle Jurassic Xintiangou Formation of China [5]. Other taxa have been described from the Middle to Late Jurassic of Sichuan, China: *Agilisaurus louderbacki*, *Hexinlusaurus multidens*, and *Xiaosaurus dashanpensis* from the Lower Shaximiao Formation, as well as *Yandusaurus hongheensis* from the Upper Shaximiao Formation [1,9–12]. Basal neornithischians from Asia also include *Kulindadromeus zabaikalicus* from Siberia in Russia [13], which bore feather-like structures.

In Southeast Asia, more recently diverging neornithischians have been described, including iguanodontian ornithopods and basal ceratopsians such as *Mandschurosaurus laosensis* from the Grès Supérieurs Formation of Laos [14]; *Siamodon nimngami* [15], *Ratchasimasaurus suranareae* [16], *Sirindhorna khoratensis* [17], and *Psittacosaurus sattayaraki* [18] from the Khok Kruat Formation of Thailand; all of these cerapodan neornithischians (ornithopods and marginocephalians) are Early Cretaceous (Aptian-Albian) in age. However, no valid taxon had been described yet from an older formation. A few elements of basal neornithischians were reported from the Late Jurassic–Early Cretaceous Phu Kradung

Formation of Thailand including a left femur from the Dan Luang locality, Mukdahan Province [19,20], and a lower jaw with the distinct characters of fan-shaped teeth and asymmetrically distributed enamel on the lingual surface from the Phu Noi locality, Kalasin Province [21].

In 2012, a team from the Palaeontological Research and Education Centre, Mahasarakham University, excavated remains from the well-consolidated rock at the Upper Jurassic Phu Noi locality in the lower part of the Phu Kradung Formation. Laboratory preparation revealed a well-preserved articulated skeleton of a new basal neornithischian dinosaur, which is the earliest neornithischian taxon from Southeast Asia. The discovery of this dinosaur provides new information about the biodiversity, biogeography, and early evolutionary history of neornithischians during the Late Jurassic–Early Cretaceous time interval.

### 1.2. Geological Settings

The Phu Kradung Formation is considered to form the base of the Mesozoic Khorat Group and is distributed on the Khorat Plateau in northeastern Thailand. Sedimentological aspects indicate that it was deposited by meandering rivers, with a high-energy regime along palaeochannels, followed by deposition in a floodplain and lacustrine environment [22,23].

The specimen was excavated from the Phu Noi locality (Figure 1), which belongs to the lower part of the Phu Kradung Formation, which may be Late Jurassic in age [24,25]. It is embedded in a matrix formed of brownish-purple and greenish-grey sandy siltstones which were deposited in a natural levee with low energy; the elevation of the bone bed is about 243–258 m above sea level [26,27].

The Phu Noi locality is located at Ban Din Chi, Kham Muang District, Kalasin Province, Thailand. It is one of the richest Southeast Asian non-marine vertebrate bone-beds where teams from the Palaeontological Research and Education Centre, Mahasarakham University, and Sirindhorn Museum have worked for more than ten years. Several new taxa have been named from Phu Noi, including the freshwater hybodont shark *Acrodus kalasinensis* [24], the ginglymodian actinopterygian *Isanichthys lertboosi* [28], the lungfish *Ferganoceratodus annekempae* [29], the xinjiangchelyid turtles *Phunoichelys thirakupti* and *Kalasinemys prasarttongosothi* [30,31], and the teleosaurid crocodylomorphs *Indosinosuchus potamosiamensis* and *Indosinosuchus kalasinensis* [25,32]. This site has also produced brachyopid temnospondyls, pterosaurs, and numerous dinosaur elements belonging to mamenchisaurid sauropods, metriacanthosaurid (=sinraptorid) theropods, and small basal neornithischians [20,21,33–37] that are currently under study. The Phu Noi site is close to the Late Jurassic Phu Kradung Formation outcrop of Ban Khok Sanam village (about 3 km away from Phu Noi) where a dorsal vertebra of a stegosaur was reported [20,38]. This vertebrate fauna resembles that of the Middle to Late Jurassic from China.

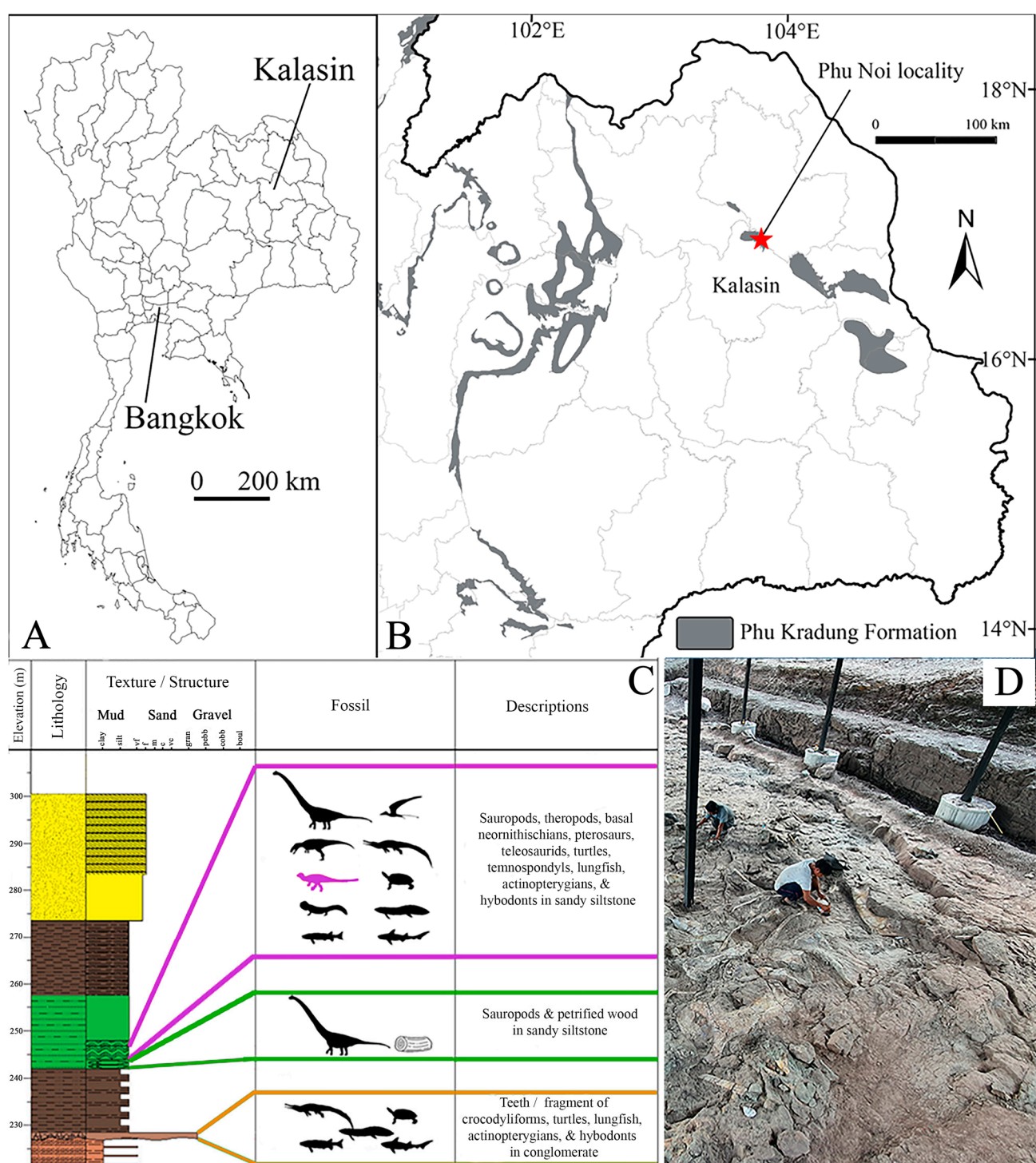

**Figure 1.** Locality map and stratigraphy of Phu Noi locality, which has yielded the holotype of *Minimocursor phunoiensis* gen. et sp. nov. Map of Thailand, showing the location of Bangkok and Kalasin Province (**A**); map of Khorat Plateau, showing the distribution of the Lower Phu Kradung Formation and the location of Phu Noi locality (red star) (**B**); stratigraphic column of Phu Noi (modified from [26]) (**C**); photograph of the excavation site in 2023 (**D**).

## 2. Materials and Methods

*2.1. Examined Material*

The material referred to *Minimocursor phunoiensis* gen. et sp. nov. consists of one partially articulated specimen (PRC 150, the holotype), a left pes with tibia (SM2021-1-132), and an isolated lower jaw (PRC 149). All this material was prepared using a pneumatic pen and consolidated with cyanoacrylate glue. Measurements of the bones were taken with a digital caliper. UV light observations were made using a UV flashlight emitting at 80 W (λ excitation centred around 365 nm).

The holotype, prepared over more than five years, consists of an articulated specimen that is more than 50% complete. It comprises postcranial elements including a well-preserved articulated series of vertebrae with ossified tendons, the left scapula with some part of the coracoid, the entire pelvic girdle, the left femur, tibia and fibula, the left tarsals, and metatarsals (Figures 2 and 3). Some displaced bones include the right jugal, the left surangular and angular, the left manus, the right femur, tibia and fibula, a phalanx, an ungual, and ribs (Figures 2 and 3). The specimen is now housed in the collection of the Palaeontological Research and Education Centre (PRC), Mahasarakham University, Thailand. Additional materials of *Minimocursor phunoiensis* gen. et sp. nov. are kept at the PRC and Sirindhorn Museum (SM), Sahatsakhan, Kalasin Province, Thailand.

*2.2. Phylogenetic Analysis*

The systematics of basal neornithischians is still problematic [1,2]. A variety of basal forms were traditionally referred to as hypsilophodontids [1–3]. Many taxa, which were once considered early members of ornithopods, have been transferred into the basal neornithischians [6,8] (see Table 1).

The phylogenetic position of *Minimocursor phunoiensis* gen. et sp. nov. is investigated in a series of three cladistic analyses using the matrix published by Sues et al. (2023 [39], itself modified after Madzia et al., 2018 [6]). This matrix initially contained 255 characters and 75 taxa including a wide range of ornithischians. *Minimocursor* gen. nov. is added and scored in one or three Operational Taxonomic Units (OTUs), depending on the analysis (see below). The genus *Marasuchus* is defined as outgroup. The analyses are conducted using TNT 1.6 [40], with all characters unordered and of equal weight. The heuristic search algorithm consists of 100 replicates of Random Stepwise Addition (RAS) and Tree–Bisection–Reconnection branch swapping (TBR), holding 10 trees per replicate. The memory buffer ("Maxtree") is set at 50,000 trees. If individual replications fill the memory buffer ("some replications overflowed"), a second round of RAS + TBR is then applied to the resulting trees ("trees from RAM"). The Bremer support of the nodes is calculated from the suboptimal trees up to 5 steps longer. The strict consensus is calculated for all the analyses.

In order to discard wildcard taxa, the first analysis includes all the taxa of the matrix of [39], in addition to the OTU *Minimocursor*_gen. nov., based on all the specimens cited in the description (i.e., holotype PRC 150, isolated dentary PRC 149 and pes SM2021-1-132). The command 'prunnelsen' of TNT is used to identify the wildcard taxa, and two OTUs (*Leaellynasaura* and *Notohypsilophodon*, see discussion for more details) are removed in the two following analyses.

The second analysis aims to test the conspecificity of the additional material attributed to *Minimocursor* gen. nov. In addition to the OTU *Minimocursor*_gen.nov., the holotype PRC 150 and the dentary PRC 149 are scored as independent OTUs (*Minimocursor*_holotype and *Minimocursor*_lower_jaw, respectively). The pes SM2021-1-132 is not used as an independent OTU owing to the very limited number of observable characters.

The third analysis examines the position of *Minimocursor* gen. nov. among Ornithischia. The genus is scored in the data matrix by the unique OTU *Minimocursor*_gen.nov.

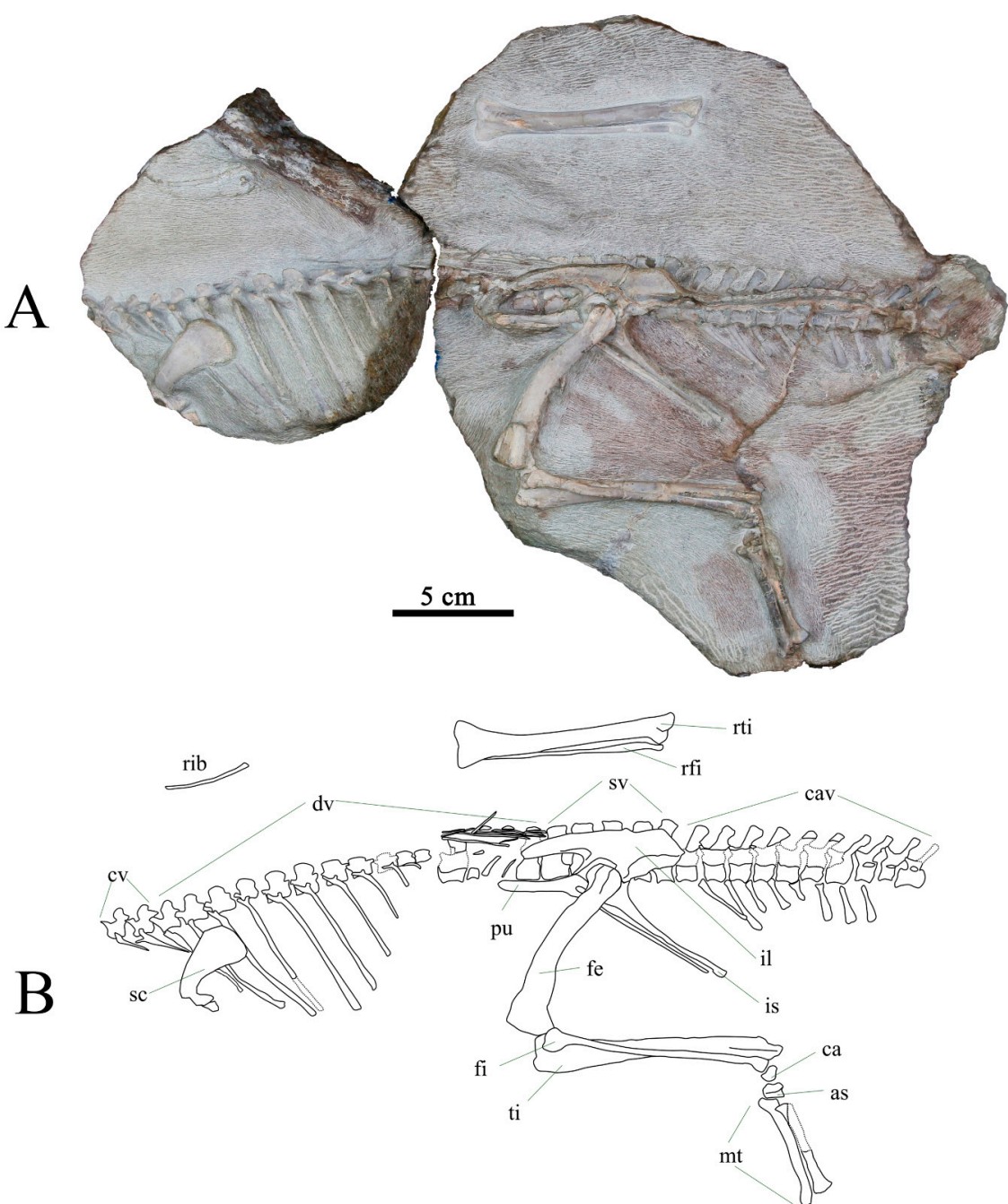

**Figure 2.** Holotype of *Minimocursor phunoiensis* gen. et sp. nov., PRC 150, in left lateral view. Photograph (**A**) and interpretative drawing (**B**).

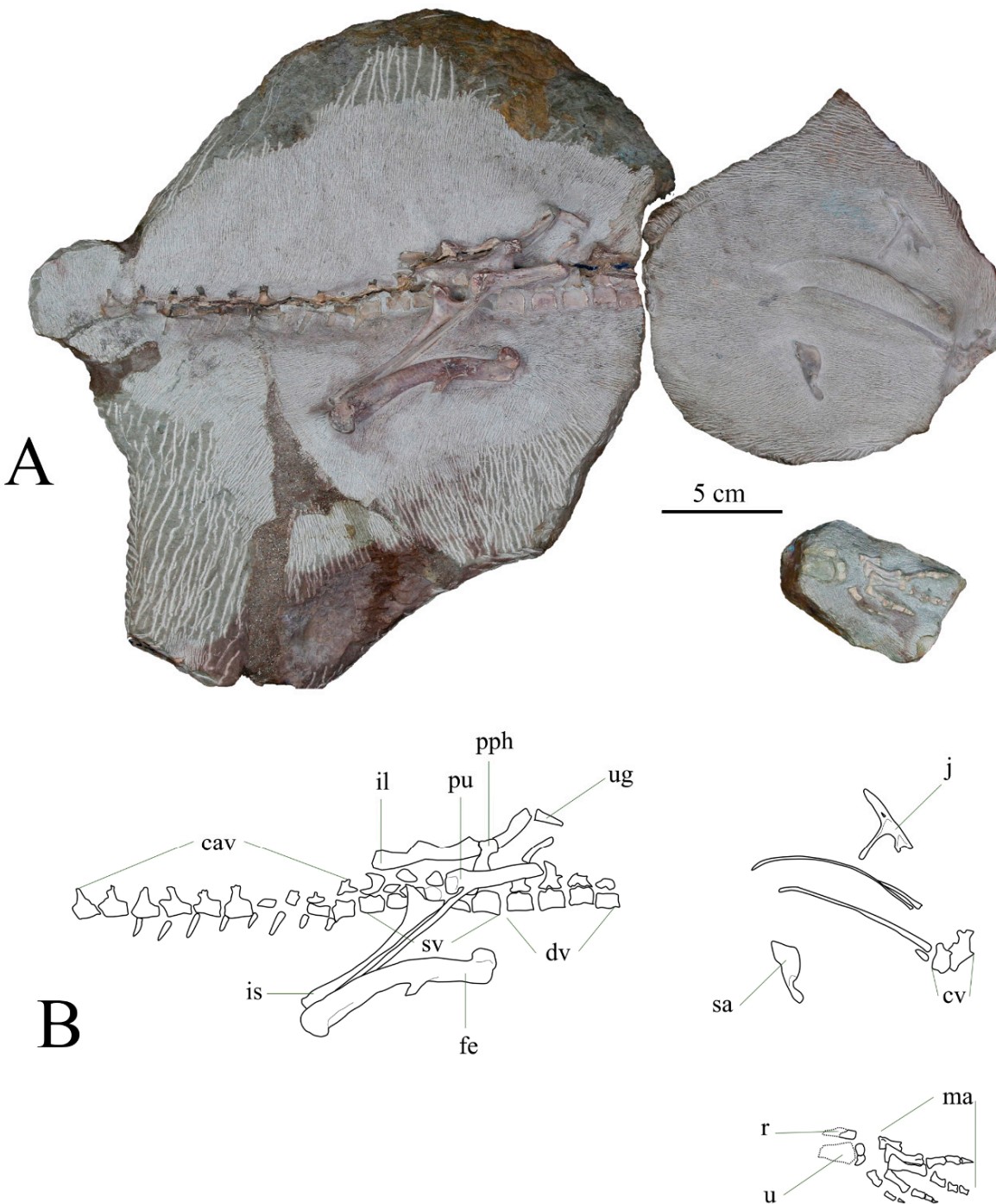

**Figure 3.** Holotype of *Minimocursor phunoiensis* gen. et sp. nov., PRC 150, in right lateral view. Photograph (**A**) and interpretative drawing (**B**).

**Table 1.** Occurrence of basal neornithischians in the Middle Jurassic–Early Cretaceous.

| Taxa | Age | Formation | Occurrence | Reference |
|---|---|---|---|---|
| *Sanxiasaurus modaoxiensis* | Aalenian–Toarcian | Xintiangou | China (Chongqing) | [5] |
| *Agilisaurus louderbacki* | Middle–Late Jurassic | Lower Shaximiao | China (Sichuan) | [12] |
| *Hexinlusaurus multidens* | Middle–Late Jurassic | Lower Shaximiao | China (Sichuan) | [10] |
| *Xiaosaurus dashanpensis* | Middle–Late Jurassic | Lower Shaximiao | China (Sichuan) | [9] |
| *Kulindadromeus zabaikalicus* | Bathonian | Ukureyskaya | Russia (Cherynyshevsky) | [13] |
| *Yandusaurus hongheensis* | Late Jurassic | Upper Shaximiao | China (Sichuan) | [41] |
| ***Minimocursor phunoiensis* gen. et sp. nov.** | **Late Jurassic** | **Phu Kradung** | **Thailand (Kalasin)** | **This study** |
| *Nanosaurus agilis* | Kimmeridgian–Tithonian | Morrison | USA (Colorado, Wyoming) | [42] |
| *Hypsilophodon foxii* | Barremian | Wessex | England (Isle of Wight) | [43,44] |
| *Changmiania liaoningensis* | Barremian | Yixian | China (Liaoning) | [45] |
| *Jeholosaurus shangyuanensis* | Barremian–Aptian | Yixian | China (Liaoning) | [46,47] |
| *Changchunsaurus parvus* | Aptian–Cenomanian | Quantou | China (Jilin) | [48,49] |
| *Leaellynasaura micagraphica* | Albian | Eumeralla | Australia (Victoria) | [50] |

Character scoring for *Minimocursor phunoiensis* gen. et sp. nov. based on the matrix published by Sues et al., 2023 [39]:

*Minimocursor*_gen. nov. (all the specimens including holotype PRC 150, isolated dentary PRC 149, and pes SM2021-1-132)

????? ????? ????? ????? ????? ?0??? ?0?1? ??2?? ?????
????? ????? ????? ????? ????? ??130 11011 10?00 0????
????? ?1??? ????? ????? ??01? ??0?? ??10? ????1 ??00?
0?00? ????? 01100 011?? 0000? ????? ????? ??00? ?00?0
00010 01011 11?20 0211? 00??0 ?0001 11000 12120 00-00
01221 01000 0?1?? ??0?? 1000? 1100?

*Minimocursor*_holotype (holotype PRC 150)

????? ????? ????? ????? ????? ?0??? ?0?1? ??2?? ?????
????? ????? ????? ????? ????? ????? ????? ???00 0????
????? ?1??? ????? ????? ????? ????? ????? ????? ?????
????? ????? 01100 011?? 0000? ????? ????? ??00? ?00?0
00010 01011 11?20 0211? 00?00 ?0001 11000 12120 00-00
01221 01000 0?1?? ????? ????? 1100?

*Minimocursor*_lower_jaw (isolated dentary PRC 149)

????? ????? ????? ????? ????? ????? ????? ????? ?????
????? ????? ????? ????? ????? ??130 11011 11??? ?????
????? ????? ????? ????? ??01? ??0?? ??10? ????1 ??00?
0?00? ????? ????? ????? ????? ????? ????? ????? ?????
????? ????? ????? ????? ????? ????? ????? ????? ?????
????? ????? ????? ????? ????? ?????

*2.3. Anatomical Terminologies*

a, angular; ac, acetabulum; ag, alveolar groove; aofo, antorbital fossa; ap, acromion process; ar, articular; as, astragalus; ap, anterior process (of jugal); bs, brevis shelf; c, coronoid process; ca, calcaneum; cav, caudal vertebrae; caig, caudal intercondylar groove; cf, caput femoris (femoral head); cmc, proximal caudomedial condyle of tibia; cnc, cnemial crest; cr, caudal rib (transverse processes or pleurapophyses); co, coracoid; cr, cervical rib; cv, cervical vertebrae; d, dentary; def, deltoid fossa; der, deltoid ridge; dia, diapophysis; dp, dorsal process (of jugal); dr, dorsal rib; dv, dorsal vertebrae; f, frontal; fe, femur; fi, fibula; ftr, fourth trochanter; gl, glenoid; gtr, greater trochanter; ha, haemal arch/process; i, intermedium; I-V, digits I-V; il, ilium; ilp, iliac peduncle of ischium and pubis; is, ischium; isf, distal 'foot' of the ischium; isp, ischial peduncle of ilium and pubis; itf, infratemporal fenestra; j, jugal; jb, jugal boss; l, lacrimal; lc, proximal lateral condyle of tibia; lcd, distal lateral condyle of femur; lm, lateral malleolus; ls, ligament sulcus; ltr, lesser trochanter; ma, manus; mc, metacarpal; mcd, distal medial condyle of femur; mm, medial malleolus; mt, metatarsal; mx, maxilla; na, nasal; nc, neural canal; ncs, neurocentral suture; ns, neural spine; o, orbit; ot, ossified tendon; pap, palpebral; pcdl, posterior centrodiapophyseal lamina; pd, predentary; ph, phalanx; pmx, premaxilla; po, postorbital; poc, paroccipital process; pop, postacetabular process; por, posterior pubic ramus; poz, postzygapophysis; pp, posterior process (of jugal); ppb, prepubic process; pph, pedal phalanx; ppn, neck of the prepubic process; prf, prefrontal; prp, preacetabular process; prz, prezygapophysis; pu, pubis; pup, pubic peduncle of ilium and ischium; q, quadrate; qj, quadratojugal; r, radius; ra, radiale; rap, retroarticular process; rfi, right fibula; rti, right tibia; s, symphysis; sa, surangular; saf, supraacetabular flange; sap, supracetabular process; sc, scapula; scb, scapular blade; scl, scapular labrum; sp, spinal process; spp, scapular proximal plate; sq, squamosal; sr, sacral rib; sv, sacral vertebrae; ti, tibia; tub, tubercle; u, ulna; ug, ungual; ul, ulnare.

*2.4. Institutional Abbreviations*

**PRC**, Palaeontological Research and Education Centre, Mahasarakham University, Maha Sarakham, Thailand; **SM**, Sirindhorn Museum, Sahatsakhan, Thailand.

## 3. Results

*3.1. Systematic Palaeontology*

Dinosauria Owen, 1842 [51]
Ornithischia Seeley, 1888 [52]
Neornithischia Cooper, 1985 [53]
*Minimocursor phunoiensis* gen. et sp. nov.

### *Minimocursor* gen. nov.

*ZooBank LSID*: urn:lsid:zoobank.org:act:F9A272CA-5A93-4E01-A5A1-93671D5FF3E3
*Type species*. Minimocursor phunoiensis gen. et sp. nov.; see below.
*Diagnosis*. As for the type and only species, *Minimocursor phunoiensis* (see below).

### *Minimocursor phunoiensis* gen. et sp. nov.

*ZooBank LSID*: urn:lsid:zoobank.org:act:6B1DF01F-0D3E-4837-99B2-3D358CA2F928
*Holotype*. PRC 150 (Figures 2–4). Partially articulated skeleton comprising a series of vertebrae (from the last three cervical to the 10th caudal) with a few ossified tendons; left scapula and manus; entire pelvic girdle; left femur, tibia, and fibula; left tarsals and metatarsals; and a few displaced bones: right jugal, left surangular and angular, incomplete tooth (now considered lost), right femur, tibia and fibula, phalanx, and pedal ungual.

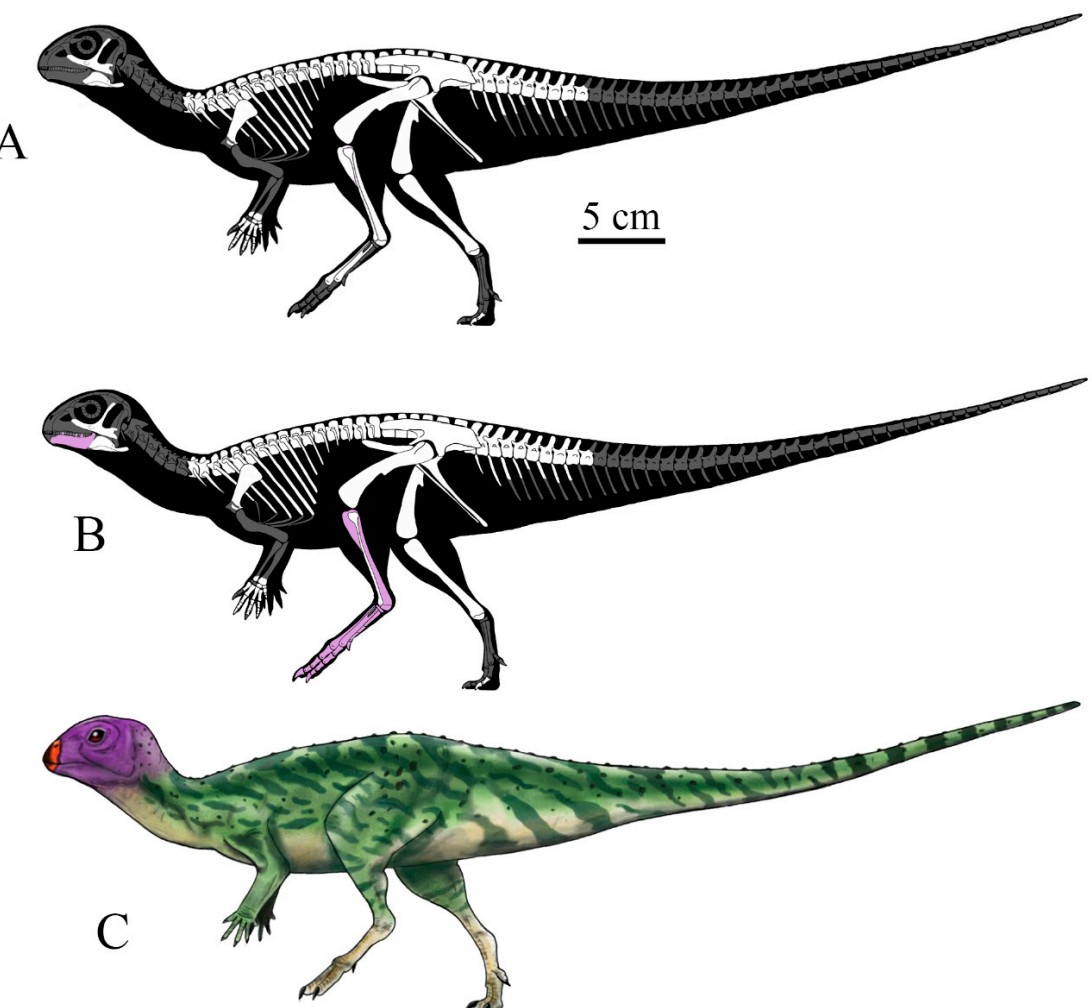

**Figure 4.** Reconstruction of *Minimocursor phunoiensis* gen. et sp. nov. (PRC 150) in left lateral view (except reversed images of the right jugal). Recovered elements of the holotype shown in white (**A**); holotype with referred materials, which are not to scale, shown in light purple (**B**); and life restoration (**C**). Drawings by Wongwech Chowchuvech (**A**,**B**) and Sita Manitkoon (**B**).

*Referred material*. PRC 149 (renumbered from PN 13-09 in Buffetaut et al., 2014 [20,21], isolated lower jaw; SM2021-1-132, left pes with tibia. All from the type locality.

*Diagnosis*. Ornithischian dinosaur distinguished by a unique combination of plesiomorphic and apomorphic characters resembling those of basal neornithischians: jugal posterior process bifurcated distally (present in *Lesothosaurus*, *Jeholosaurus*, *Psittacosaurus*, and some early thyreophorans including *Emausaurus* and *Scelidosaurus*); a low flattened subtriangular boss projects laterally on the surface of the jugal (report in *Changchunsaurus*, and some early ornithischians including *Manidens* and *Lioceratops*); the pre-caudal axial skeleton is composed of 15 dorsals and 5 sacrals (presente in *Lesothosaurus*, *Agilisaurus* and *Hexinlusaurus*); the brevis shelf of the ilium is visible in lateral view along its entire length (present in *Agilisaurus*, *Sanxiasaurus* and *Lesothosaurus*); a distinct supraacetabular flange is present on the pubic peduncle of the ilium (present in *Agilisaurus*, *Sanxiasaurus*, and some early thyreophorans including *Scutellosaurus* and *Scelidosaurus*).

*Type locality and horizon*. Phu Noi locality (a small hill as the Thai name indicates), Ban Din Chi Sub-district, Kham Muang District, Kalasin Province, Thailand; Upper Jurassic Phu Kradung Formation, Khorat Group. This locality is part of the Kalasin Geopark area.

*Etymology*. The generic name is from the Latin '*minimus*', 'the smallest', referring to the holotype individual, which is smaller than any other neornithischians from the site,

combined with the suffix '*-cursor*', the Latin word for runner. The specific epithet is derived from the excavation site, Phu Noi.

*3.2. Anatomical Description*

3.2.1. Skull and Mandible

*Jugal.* The isolated right jugal is preserved in a lateral view (Figure 5). The anterior process of the jugal is straight as in many ornithischians, but not anterodorsally inclined as in *Agilisaurus* [12]. The dorsal process is slightly inclined posteriorly to overlie dorsally with the ventral process of the postorbital. The posterior process is a laterally compressed thin plate that extends bifurcated anterodorsally to contact with the dorsal quadrate. Between the anterior and dorsal processes, a low jugal boss projects laterally on the surface of the bone below the orbit. It has a flattened subtriangular shape. A jugal boss, or jugal horn, is present in some heterodontosaurids such as *Heterodontosaurus* and *Manidens* [54,55], and most marginocephalians. In non-cerapodan neornithischians, the jugal boss has been reported only in some Cretaceous thescelosaurids including *Changchunsaurus* [48,56], *Zephyrosaurus* [57,58], and *Orodromeus* [59]. The low boss of *Changchunsaurus* is described as a nubble structure [48] or a node-like rugose surface texture [56] contrary to the jugal boss of *Zephyrosaurus* and *Orodromeus*, which is a high, posteriorly projected boss on the caudolateral surface of the jugal [56–59].

*Surangular.* The left surangular and angular are exposed in the lateral view (Figure 5). The surangular forms the caudal half of the coronoid process of the dentary. The surangular foramen cannot be observed in *Minimocursor* gen. et sp. nov. as in most basal neornithischians, but a small foramen can be seen in some ornithischians such as *Heterodontosaurus*, *Hypsilophodon*, *Thescelosaurus*, and *Gasparinisaura* [60]. The surangular forms a long caudoventral process through having the articular and the retroarticular process for contact with the quadrate.

*Angular.* The angular closely overlaps the ventral surface of the surangular in the lateral view, and confers a wedge shape to the posteroventral portion of the mandible [60] (Figure 5). Its lateral surface is dorsoventrally convex. The ventral margin of the angular is slightly straight to slightly convex as in *Hypsilophodon* [43], *Jeholosaurus* [61], *Changchunsaurus* [56], and *Changmiania* [45], contrary to the slightly concave margin in *Haya* [62,63].

*Tooth.* A single isolated tooth was found embedded in the matrix of the holotype specimen but is now lost (Figure 6). The crown is broken. The cylindrical root is straight in lateral view.

3.2.2. Axial Skeleton

*Cervical vertebrae.* The last three cervical vertebrae are preserved and articulated (Figure 7), but the first one is only formed of the imprint of the neural spine and postzygapophyses. The centra can be observed in the right lateral view. Even though the anterodorsal margin of the second vertebra is broken, the cervical centra have an almost rectangular outline, longer than high as in *Hexinlusaurus* [10], *Hypsilophodon* [43], and *Koreanosaurus* [64] (Figure 7). The lateral surfaces of the centra are concave anteroposteriorly. The diapophyses form a prominent process on the lateral side of the neural arch and project posterolaterally, forming an angle of approximately 45° with the horizontal plane in both lateral and anterior views. The neural spines are rectangular in shape in lateral view, located more dorsally on the neural arch, with the apex anteroposteriorly expanded. The prezygapophyses face dorsolaterally and have facets oriented at an angle of approximately 45° from horizontal, whereas the postzygapophyses point straight horizontally. The articular facets of the pre- and postzygapophyses are flat. The postzygapophyses are short and slightly higher than the prezygapophyses. The cervical ribs are short, double-headed, and gradually narrow distally.

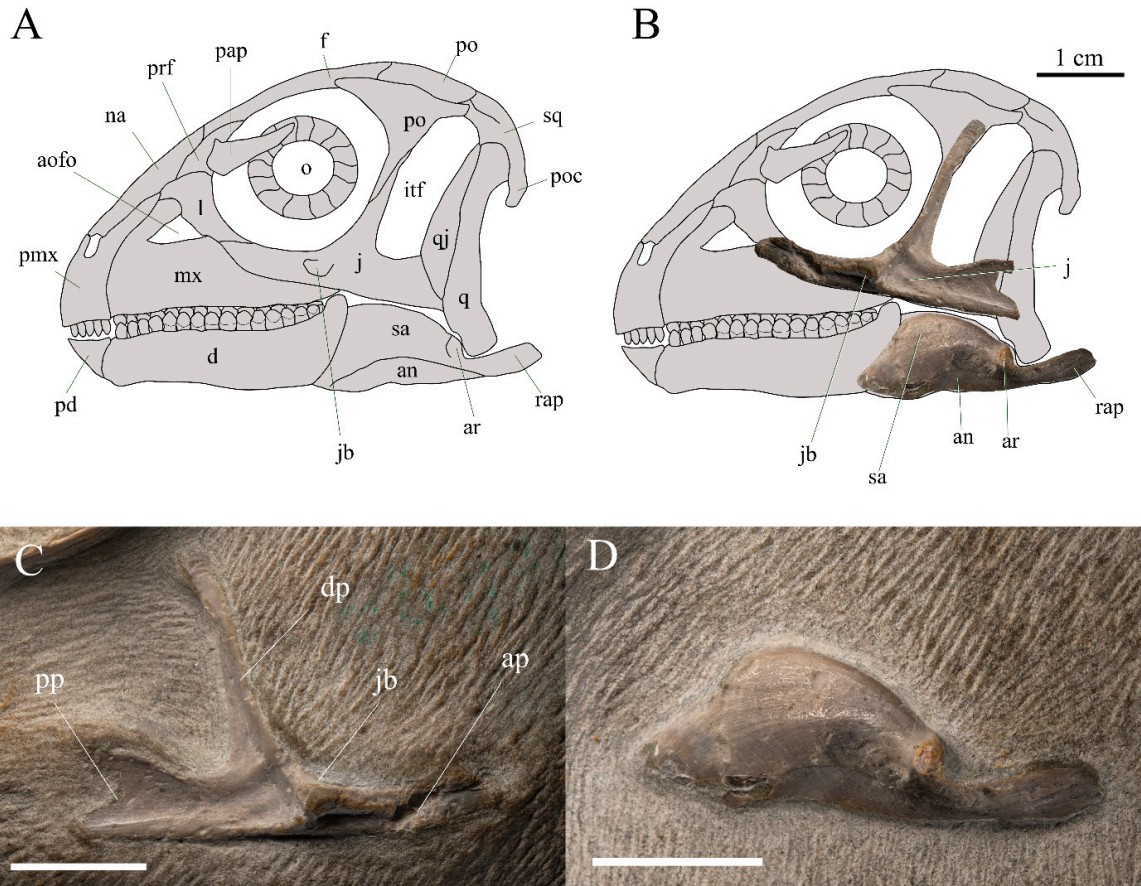

**Figure 5.** Skull of *Minimocursor phunoiensis* gen. et sp. nov. based on PRC 150 in left lateral view. Outline reconstruction of the skull (**A**); outline reconstruction of the skull with photos of the jugal (reversed), surangular, and angular (**B**); right jugal (**C**); and left surangular and angular (**D**). Scale bars: 1 cm.

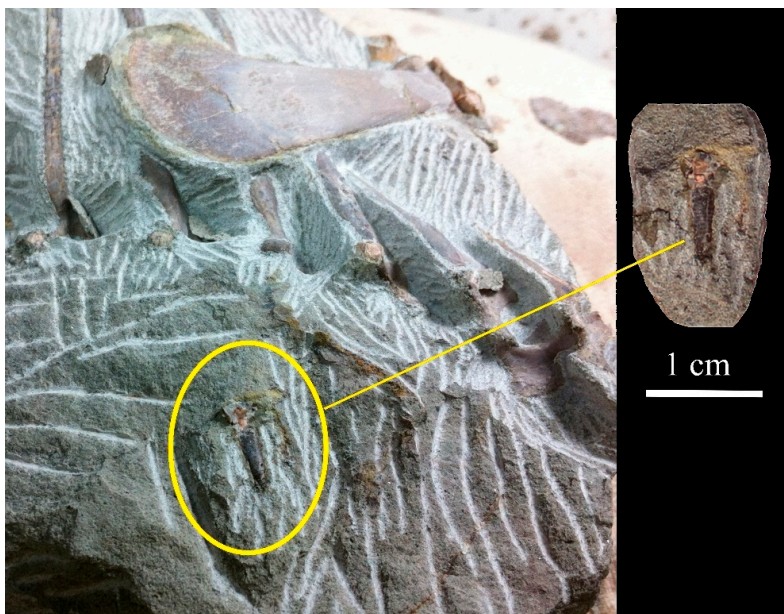

**Figure 6.** Isolated tooth of *Minimocursor phunoiensis* gen. et sp. nov., PRC 150, now lost.

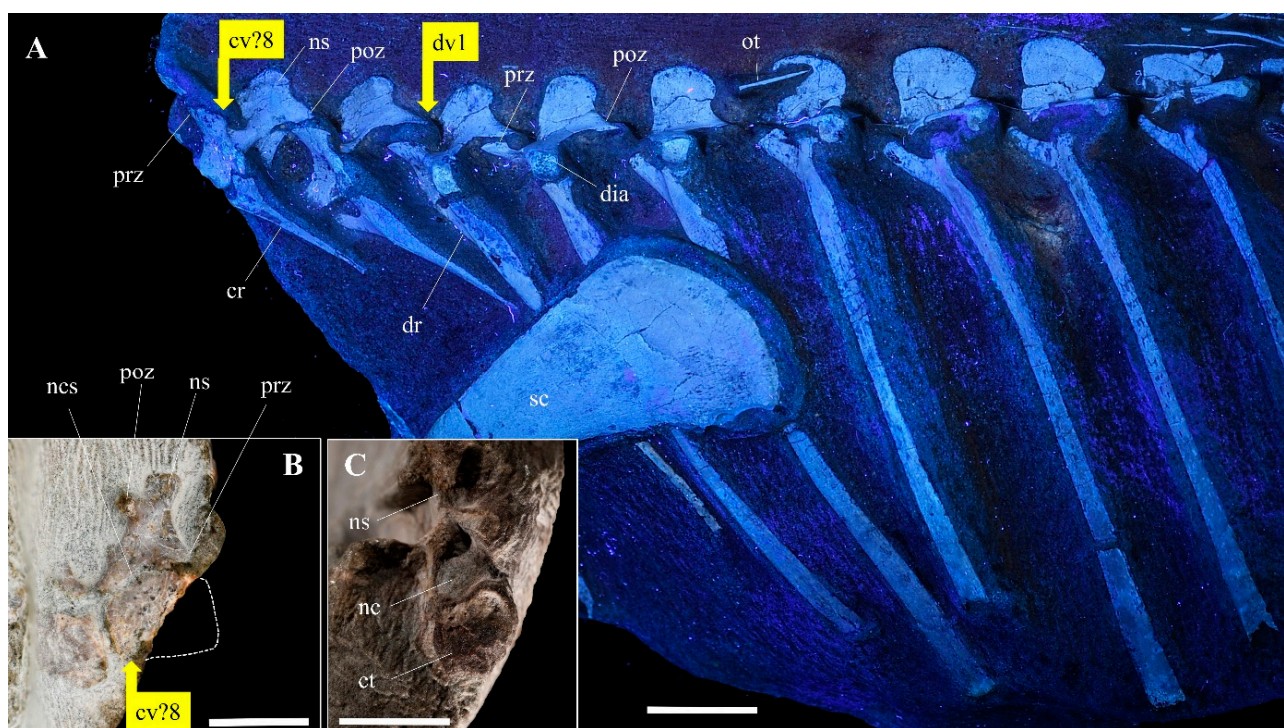

**Figure 7.** Distal cervical and proximal dorsal vertebrae of *Minimocursor phunoiensis* gen. et sp. nov., PRC 150, in left lateral view, under UV light (**A**); distal cervical vertebrae in right lateral view (**B**); dorsal vertebra 12 in anterior view (**C**).

**Table 2.** Vertebrae count of some ornithischian dinosaurs.

| Taxa | Cervical | Dorsal | Sacral | Caudal | Reference |
|---|---|---|---|---|---|
| *Heterodontosaurus tucki* | 9 | 12 | 6 | 34–37 | [65] |
| *Lesothosaurus diagnosticus* | 9 | 15 | 5 | 35–50 | [66] |
| *Agilisaurus louderbacki* | 9 | 15 | 5 | 44 | [12] |
| *Hexinlusaurus multidens* | 9 | 15 | 5 | 14+ | [10] |
| ***Minimocursor phunoiensis*** | **?** | **15** | **5** | **10+** | **This study** |
| *Nanosaurus agilis* | 9 | 15 | 6 | ? | [67] |
| *Hypsilophodon foxii* | 9 | 15–16 | 5–6 | 45–50 | [43] |
| *Orodromeus makelai* | 9 | 15 | 6 | 30+ | [59] |
| *Jeholosaurus shangyuanensis* | 9 | 15 | 6 | 15+ | [46,47] |
| *Changchunsaurus parvus* | 9 | 15 | ? | ? | [48,49] |
| *Changmiania liaoningensis* | 6 | 15–16 | ? | 36+ | [45] |
| *Haya griva* | 9 | ?15 | 6 | 19+ | [62,63] |
| *Convolosaurus marri* | 9 | 15 | 6 | 43 | [68] |

*Dorsal vertebrae.* The complete series of dorsal vertebrae is visible on the lateral side of the holotype (Figures 2, 3 and 7). It includes 15 articulated vertebrae (Figures 2, 3 and 7; Table 2) and all the associated ribs. The centra, observed only in dv12-dv15, are rectangular in outline in lateral view as in *Agilisaurus*, *Hexinlusaurus*, and *Yandusaurus* [10,12]. The anterior end of dv12 is somewhat concave (Figure 7C). A distinct suture line separating the neural arch from the centrum is clearly seen in dv12-dv15 (Figure 8), showing that these two elements were not fused; this indicates that the holotype was a skeletally immature individual.

The dorsals have striated rims around their anterolateral and posterolateral borders, as in some neornithischians including *Hexinlusaurus*, *Jeholosaurus*, *Yueosaurus*, *Orodromeus*, *Haya*, *Changchunsaurus*, *Parksosaurus*, *Thescelosaurus*, and *Hypsilophodon* [10,43,46,49,59,63,69,70] The diapophyses are horizontally extended, and slightly inclined dorsally. Parapophyses are possibly still covered by sediment. The neural spines of dv1-dv15 are thin rectangular plates. The prezygapophyses point anterodorsally at an angle of about 45° from horizontal in lateral view, whereas the postzygapophyses point straight horizontally. Articular facets of the pre- and postzygapophyses are flat. The anterior dorsal ribs are relatively long, while the posterior ribs gradually shorten. No ossification of the sternal segments of the dorsal ribs are preserved [71].

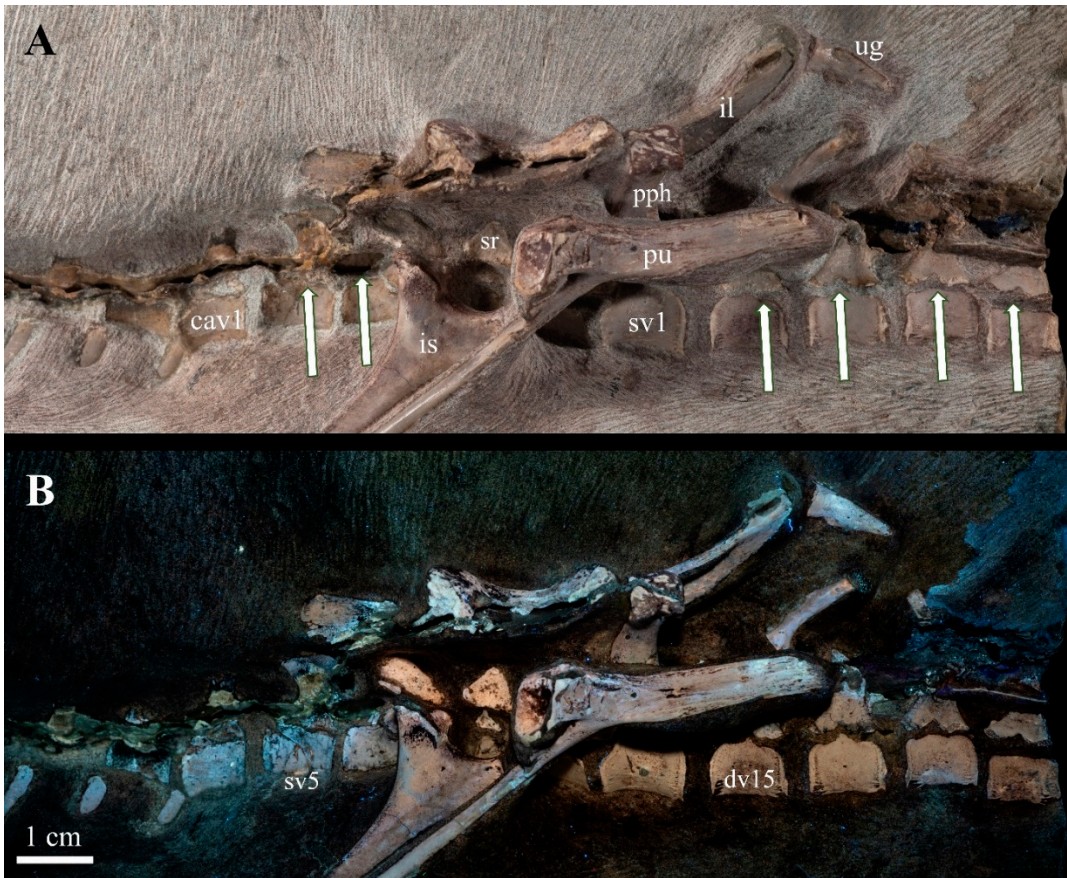

**Figure 8.** Unfused neural arch and centrum of *Minimocursor phunoiensis* gen. et sp. nov., PRC 150, in right lateral view, white arrows point the level of a suture line between the neural arch and the centrum. Photograph (**A**), under UV light (**B**).

*Sacral vertebrae.* Five sacral vertebrae are present (see Table 2). As for the dorsal vertebrae, distinct suture lines separate the neural arch from the centrum on sv1-2 (Figure 8). The centra have an almost rectangular outline, and their lateral surfaces are concave anteroposteriorly. Most of the neural arches are still buried in the matrix. The neural spines are rectangular plates and seem relatively low.

*Caudal vertebrae.* The holotype preserves 10 articulated anterior caudal vertebrae (Figures 2, 3 and 9). The shape of the centra is unclear. The neural spines are narrow, posteriorly inclined, and extend beyond their own centrum. The neural spines gradually lower posteriad, with a gradual decrease in angle to the centrum as in *Agilisaurus* and *Hexinlusaurus* [10,12]. The caudal ribs of anterior caudals are long but gradually shorten; they are horizontally extended and shaped like a long thin plate. The caudal ribs are positioned along the neurocentral suture. Haemal arches 1–8 are preserved in position, and

there is some slight damage to the distal end of arches 5, 6, and 8. Haemal arches 1–2 are the longest with a laterally flattened and slightly straightened shaft. Posteriorly, the haemal arches (ha6–8) gradually decrease in size, the shafts become flattened and expanded from the mid to the distal ends.

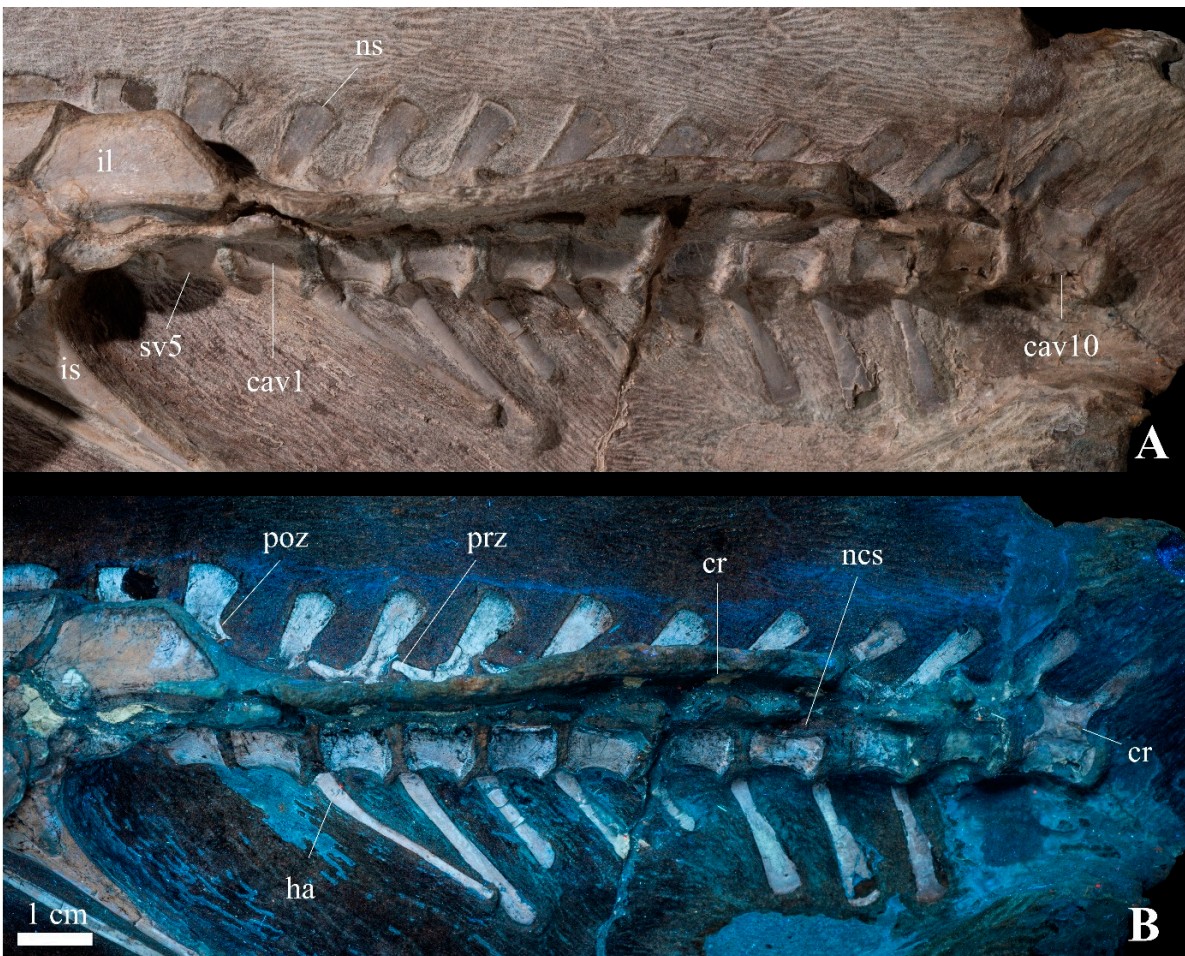

**Figure 9.** Anterior to middle caudal vertebrae of *Minimocursor phunoiensis* gen. et sp. nov., PRC 150. Photograph (**A**), under UV light (**B**).

*Ossified tendons.* The ossified tendons confirm that the specimen belongs to an ornithischian. This is the first report of ossified tendons in Southeast Asian dinosaurs. Ossified tendons run longitudinally along the lateral surfaces of the neural spines of the dorsal vertebrae (they are clearly visible from the dv4) and terminate at the first sacral vertebra (Figures 2, 7 and 10). They are long, slender, round in cross section, and appear to be parallel to each other. This type of ossified tendon is similar to those of the basal neornithischians *Agilisaurus louderbacki, Jeholosaurus shangyuanensis, Haya griva*, and some basal ceratopsians such as *Ischioceratops zhuchengensis* [12,46,62,72].

### 3.2.3. Pectoral Girdle and Forelimb

Only the left scapula, some fragmentary part of coracoid, and manus are preserved in the holotype of *Minimocursor phunoiensis* gen. et sp. nov.

*Scapula.* The left scapula is oriented at an angle of approximately 45° from horizontal (Figure 10); it is missing the acromion process and most part of the proximal plate (Figure 11). The scapular blade is rather broad and short with a relatively straight dorsal margin and a dorsoventrally convex lateral surface. The distal end of the blade is relatively thin, attached to the rib cage of dv1–3 (Figure 11). Proximally, the scapular blade expands

dorsoventrally to form the proximal plate for articulation with the coracoid. The anterodorsal margin of the proximal plate is broken, but the deltoid fossa can be observed. This fossa is developed on the proximal plate, being limited by the deltoid ridge [45]. The ventral margin of the scapula forms a subtriangular spur, the scapular labrum, for supporting the glenoid.

The scapula of *Minimocursor* gen. nov. is distinctly different from the narrow and strap-like scapular blade of *Heterodontosaurus* [54], but similar to other basal neornithischians such as *Agilisaurus* [12], *Hexinlusaurus* [10], *Changmiania* [45], *Hypsilophodon* [43], *Changchunsaurus* [49], *Haya* [62,63], *Jeholosaurus* [46], and *Yueosaurus* [69]. The acromion process is well developed as that of *Orodromeus* [59] and *Oryctodromeus* [73] and is absent in *Parksosaurus* [39]. However, unpreserved anterodorsally portion of the scapula in *Minimocursor* gen. nov. prevents additional comparisons with these taxa.

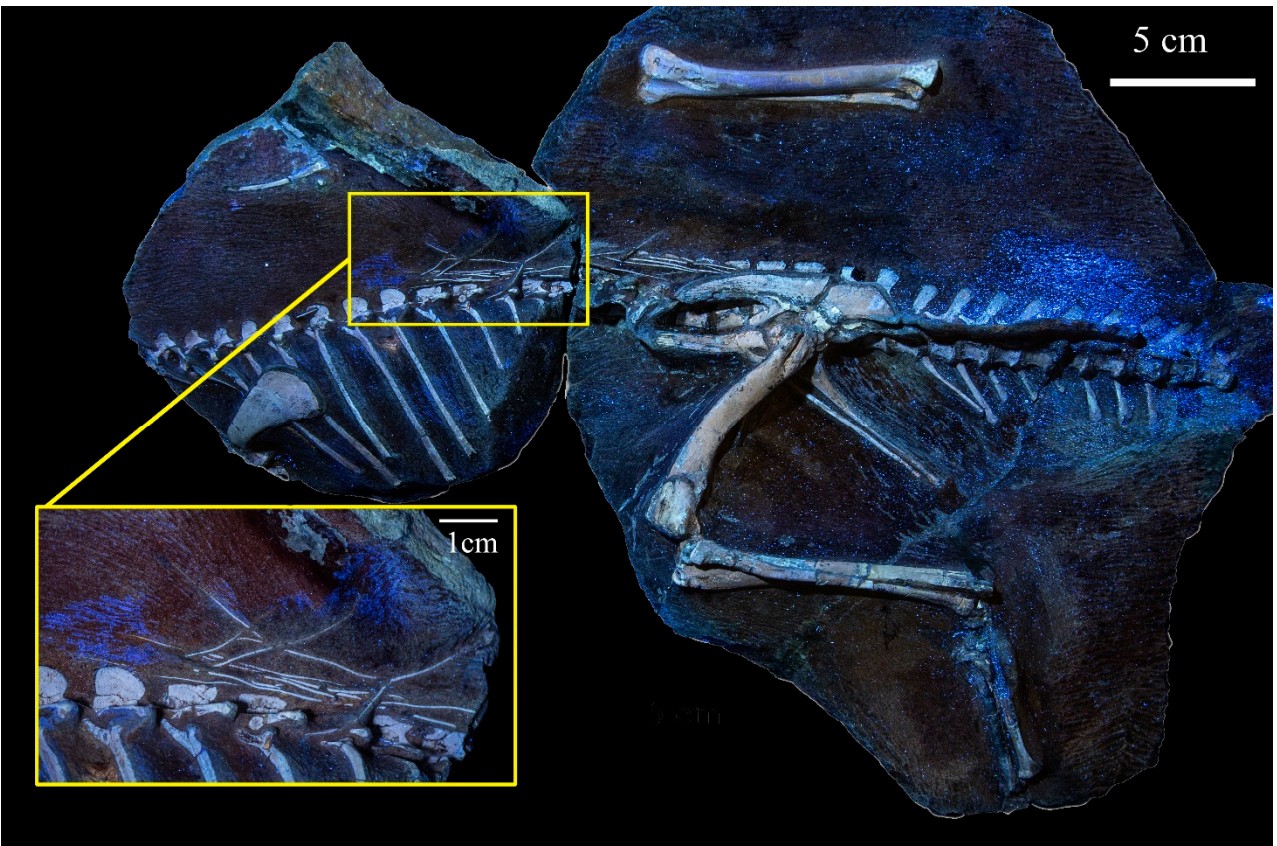

**Figure 10.** Articulated dorsal vertebrae of *Minimocursor phunoiensis* gen. et sp. nov., PRC 150, with close-up on the ossified tendons, under UV light.

Coracoid. The glenoid surface of the coracoid is partially preserved. The glenoid cavity is slightly concave dorsoventrally and has a 'D'-shaped outline.

Manus. The right forelimb is completely lost, but the left manus is articulated and preserved with relatively complete digits and carpals in anterior view (Figure 12). The trace of the distal end of the ulna and the damaged radius are still visible on the holotype. Only the ulnare and intermedium, which are subrectangular in shape, are preserved from the carpus. The ulnare articulates proximally with the distal margin of the ulna, close to the Mc V. The intermedium, which is slightly larger and more concave, articulates with the ulnare, and the medial margin of the distal radius. The carpus is not well known in basal neornithischians [60]. In *Hexinlusaurus*, it is formed of an ulnare, an intermedium, and a radiale [10]. *Hypsilophodon* bears only the ulnare, intermedium, and one distal carpal [74]. There is an ulnare, an intermedium, and a single distal carpal in *Orodromeus* [59].

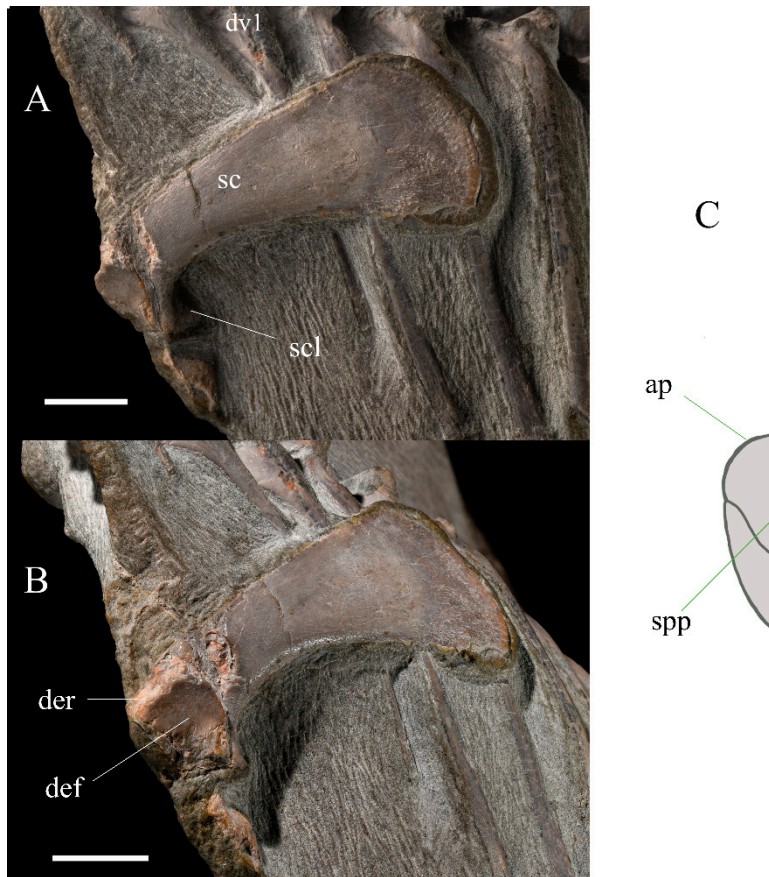

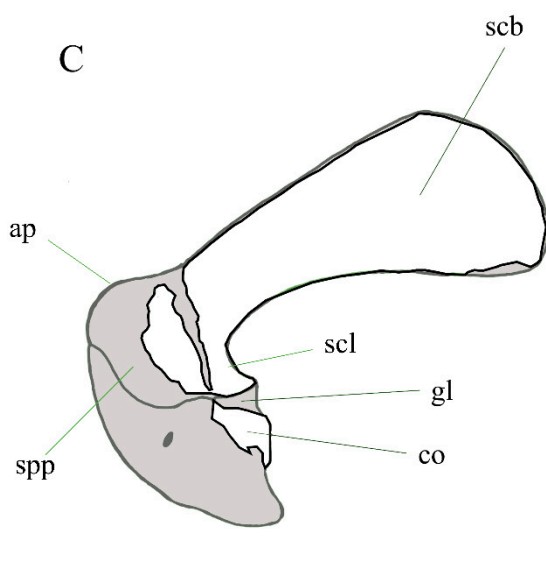

**Figure 11.** Pectoral girdle of *Minimocursor phunoiensis* gen. et sp. nov., PRC 150 in lateral view (**A**) and anterolateral view (**B**); interpretative drawing of A with estimated reconstruction of the left scapula and coracoid in grey (**C**). Scale bars: 1 cm.

The digit formula is ?-3-4-3-2 (see Table 3). Mc II and Mc III are the longest, nearly equal in length, and have greatly expanded proximal ends and constricted shafts as in other basal neornithischians such as *Hexinlusaurus* and *Hypsilophodon* [10,74]. Mc IV and Mc V are thin and flat. Mc V is the shortest metacarpal and is followed by two phalanges contrary to *Hypsilophodon* [74]. Ungual phalanges of digits I, II, and IV are clawed, but damaged in digit III.

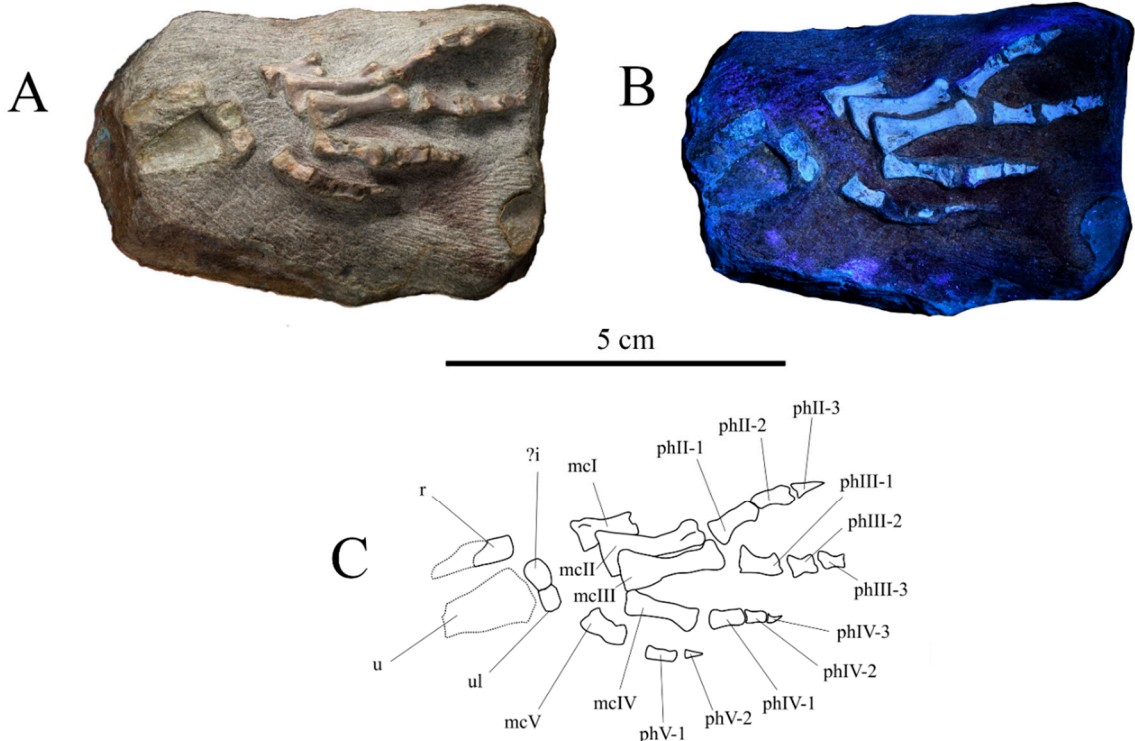

**Figure 12.** Anterior view of the left manus of *Minimocursor phunoiensis* gen. et sp. nov., PRC 150. Photograph (**A**), under UV light (**B**), and interpretative drawing (**C**).

**Table 3.** Manus and pes formula of ornithischian dinosaurs.

| Taxa | Family/Clade | Age | Manus | Pes | Source |
|---|---|---|---|---|---|
| *Heterodontosaurus tucki* | Heterodontosauridae | Early Jurassic, | 2-3-4-3-2 | 2-3-4-5-0 | [60] |
| *Xiaosaurusdashanpuensis* | Basal neornithischia | Middle Jurassic | ? | 2?-3-4-5-0 | [9] |
| *Agilisaurus louderbacki* | Basal neornithischia | Middle Jurassic, | ? | 2-3-4-5-0 | [12] |
| *Hexinlusaurus multidens* | Basal neornithischia | Middle Jurassic | 2-3-4-2?-2 | 2-3-4-5-0 | [10] |
| ***Minimocursor phunoiensis*** | **Basal neornithischia** | **Late Jurassic** | **? -3-4-3-2** | **2-3-4-5-0** | **This study** |
| *Hypsilophodon foxii* | Basal neornithischia | Early Cretaceous, | 2-3-4-3?-1? | 2-3-4-5-0 | [74] |
| *Changmiania liaoningensis* | Basal neornithischia | Early Cretaceous | 2-3-?-?-? | 2-3-4-5-? | [45] |
| *Jeholosaurus shangyuanensis* | Jeholosauridae | Early Cretaceous | ? | 2-3-4-5-0 | [46,49] |
| *Changchunsaurus parvus* | Thescelosauridae | Early Cretaceous | ? | 2-3-4-5-? | [49] |
| *Haya griva* | Thescelosauridae | Late Cretaceous | 2?-3?-4?-?-?. | 2-3-4-5-0, | [62,63] |
| *Orodromeus makelai* | Thescelosauridae | Late Cretaceous | ?-3-4-3-? | 2-3-4-5-0 | [60] |
| *Dryosaurus altus* | Iguanodontia | Late Jurassic, | 2-3-4-3-2 | 0-3-4-5-0 | [60] |
| *Tenontosaurus tilletti* | Iguanodontia | Early Cretaceous, | 2-3-3-2-1 | 2-3-4-5-0 | [75] |
| *Camptosaurus dispar* | Iguanodontia | Late Jurassic | 2-3-3-2-1 | 2-3-4-5-0 | [76,77] |
| *Iguanodon bemissartensis* | Iguanodontidae | Early Cretaceous, | 2-3-3-2-4 | 0-3-4-5-0 | [78] |
| *Edmontosaurus regalis* | Hadrosauridae | Late Cretaceous, | 0-3-3-3-3 | 0-3-4-5-0 | [60] |

3.2.4. Pelvic Girdle and Hindlimb

The left pelvic girdle and left hind limb are relatively well-preserved and present in situ for the holotype of *Minimocursor phunoiensis* gen. et sp. nov. The right femur and tibia with fibula are disarticulated. The right lateral side of the pelvic girdle is laterally compressed.

*Ilium.* The left ilium is well-preserved (Figures 2 and 13), and the outline is similar to that of *Agilisaurus*, *Hexinlusaurus,* and some other basal neornithischians. The iliac blade is a thin plate and slightly concave laterally. The preacetabular process of the ilium is narrow, long with a perfectly rounded termination, and slightly ventrally curved. The postacetabular process is short and high. There is a breakage between the preacetabular process and postacetabular process above the ischial peduncle. The acetabulum is semicircle shaped. The dorsolateral side of the acetabulum is distinctly thickened to form a round and smooth supraacetabular flange as in *Agilisaurus* and *Sanxiasaurus* [5,12]. This supraacetabular flange was subsequently lost in neornithischians more derived than *Agilisaurus* [7]. The ischial peduncle of the ilium projects ventrally. The brevis shelf of the ilium is directed ventromedially along its entire length and is clearly visible in lateral view, as in *Agilisaurus*, *Sanxiasaurus*, *Lesothosaurus*, and basal thyreophorans such as *Scelidosaurus* [1,5,12].

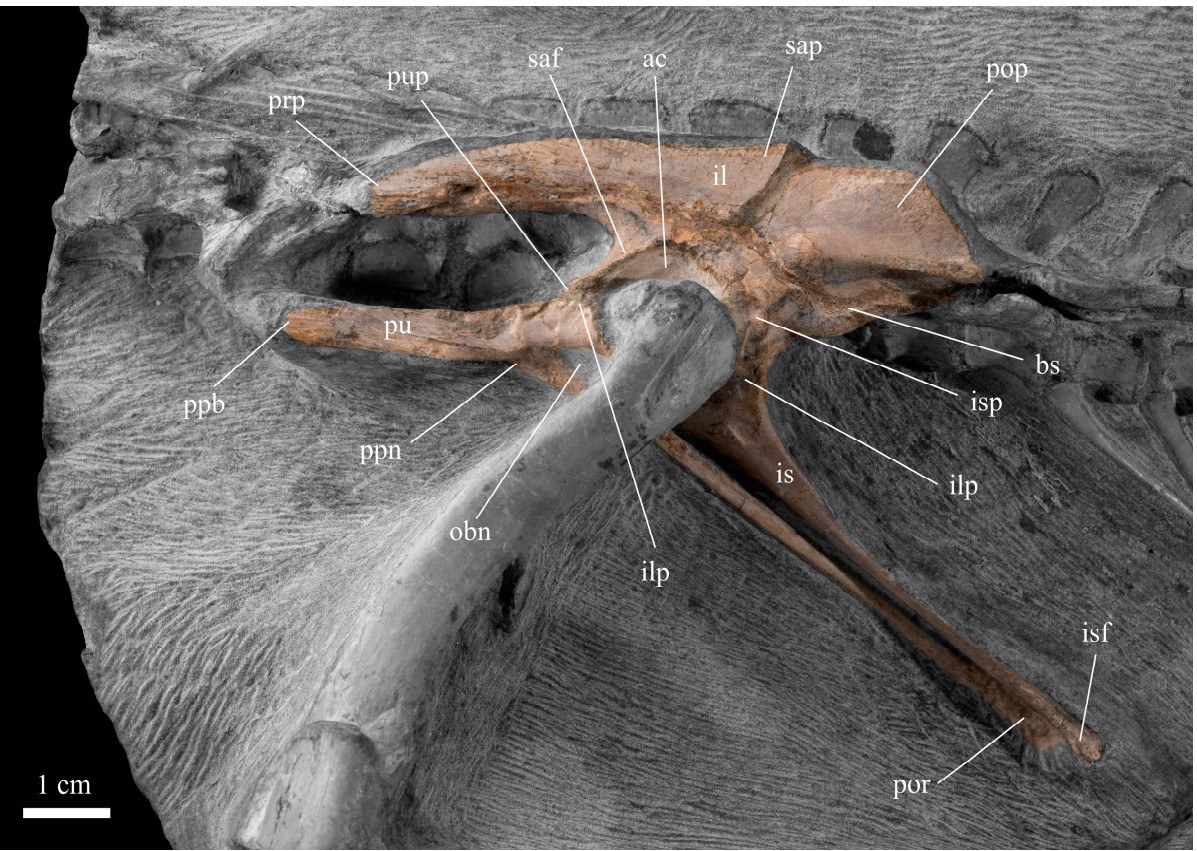

**Figure 13.** Lateral view of the left pelvic girdle of *Minimocursor phunoiensis* gen. et sp. nov., PRC 150.

*Pubis.* The left pubis is thin and long with a sharply pointed anterior end. The prepubic blade is rod-shaped, elongated into a distinct anterior process and extends beyond the distal end of the preacetabular process of the ilium as in *Hexinlusaurus* and other neornithischians, but contrary to *Agilisaurus* [10,12]. The pubis is rotated posteroventrally to lie alongside the ischium. The body of the pubis makes a substantial contribution to the margin of the acetabulum. The obturator foramen is obscured in lateral view. The posterior process is particularly elongated and terminates nearly at the same point as the ischium. The shaft is rotated at its midpoint, and the distal end is thin as in *Hexinlusaurus* [10].

*Ischium.* The ischium is long and flat with a broad proximal end for articulation with the pubis (Figures 2 and 13). The ischial shaft is relatively straight and twisted mediolaterally as in most basal neornithischians such as *Lesothosaurus, Sanxiasaurus, Agilisaurus, Hexinlusaurus, Nanosaurus* (=*Drinker, Othnielia, Othnielosaurus*), and *Kulindadromeus* [5,10,12,13,66,79]. The obturator process is still embedded in the matrix. The ischial shaft gradually broadens and becomes broadest at the distal end. There is no groove on the dorsal margin as in some basal ornithischians such as *Eocursor, Lesothosaurus,* and *Agilisaurus* [1,12,66,80].

Femur. The left and right femora are preserved (Figures 2, 3 and 14). They are robust and almost complete. The shaft of the femur is bowed in lateral view, resembling that of *Hexinlusaurus* [10], *Agilisaurus* [12], *Hypsilophodon* [81], and the *Dan Luang* neornithischian [20]. The anterior end of the greater trochanter is slightly convex, while the posterior end is strongly convex. The greater trochanter lies upon the same plane as the femoral head. The lesser trochanter is distinguished from the greater trochanter by a deep groove. The fourth trochanter is in a pendant-shaped form that lies posteromedially at one-third of the femoral length from the proximal end. A shallow fossa for muscle insertion occurs medially to the fourth trochanter. The distal portion of the femoral shaft shows a mediolateral expansion towards the distal condyles. Posteriorly, the distal condyles are separated by a caudal intercondylar groove, for the M. caudofemoralis longus [68,80], as in other basal neornithischians such as *Sanxiasaurus, Agilisaurus* and *Hexinlusaurus* [5,10,12].

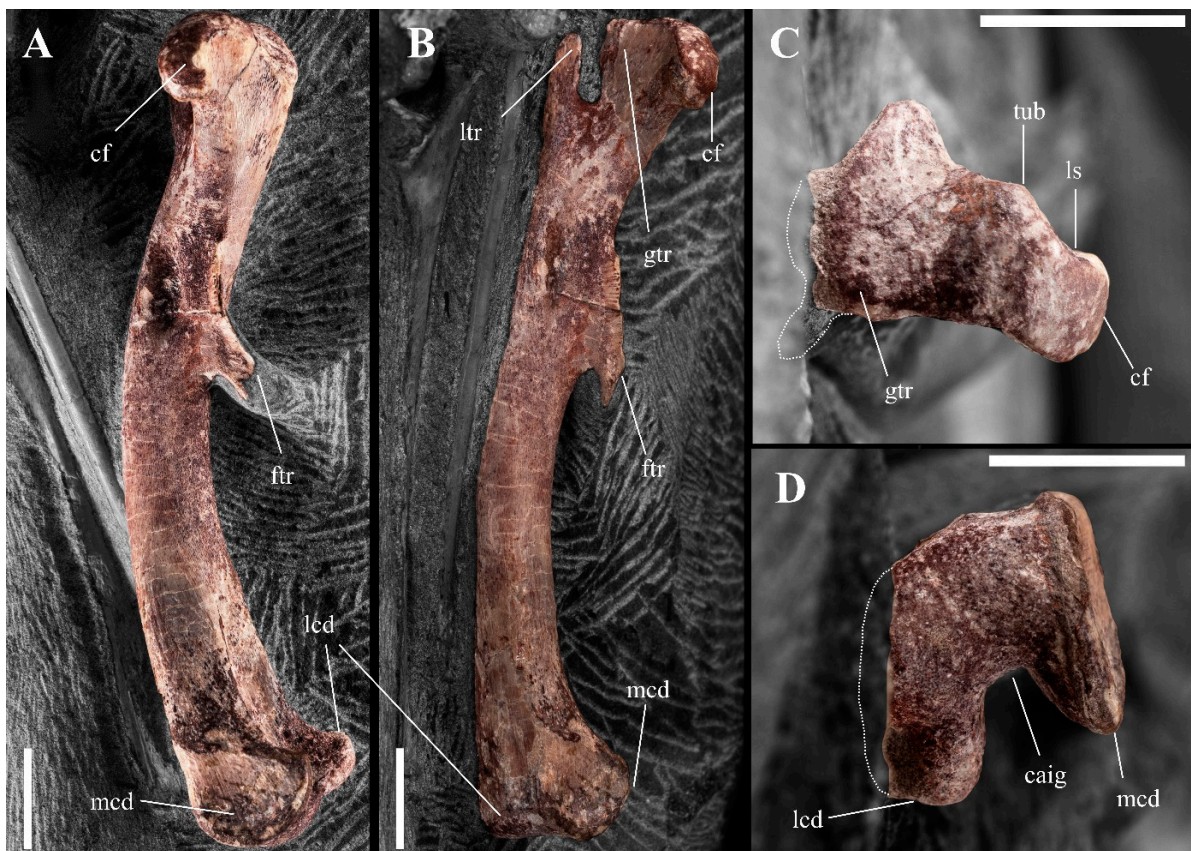

**Figure 14.** The right femur of *Minimocursor phunoiensis* gen. et sp. nov., PRC 150. Medial (**A**), anterior (**B**), proximal (**C**), and distal (**D**) views. The white dotted lines estimate the outline of the bone still covered by sediment. Scale bars: 1 cm.

*Tibia.* Both left and right tibiae are preserved together with fibulae (Figures 2 and 15). They are longer than the femora as in other basal neornithischians, indicating a fast-running adaptation [74]. The left tibia rests in lateral view while the right tibia is exposed in posterior view. The tibia shaft is robust, long, straight, rounded in transverse section, and twisted

along its long axis. Its proximal end is strongly inflated into a triangular surface and shows a division into three distinct processes: cnemial crest, fibular condyle, and medial condyle. The well-developed cnemial crest projects forward and curves laterally to form a fossa for accommodating the head of the fibula. The distal end is a distinctly transversely broadened triangle with a concave anterior surface, convex posterior surface, and a swollen ridge at its midpoint as in *Agilisaurus* [12]. The proximal tibia also contributes to the platform for the lateral femoral condyle and contributes to the base of the cnemial crest. The medial malleolus extends farther distally for accommodation of the distal end of the fibula on its anterior surface.

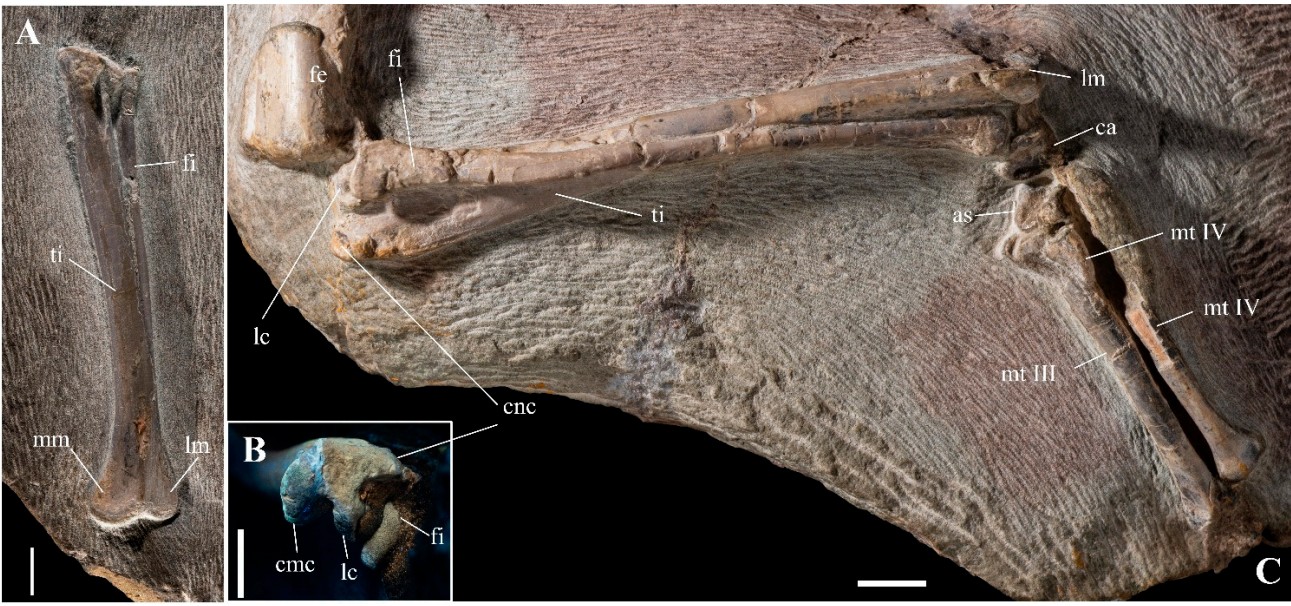

**Figure 15.** Hind limb of *Minimocursor phunoiensis* gen. et sp. nov., PRC 150. The right tibia and fibula in posterior (**A**) and proximal (**B**) views. The left tibia, fibula, and tarsals in lateral view (**C**). View (**B**) under UV light. Scale bars: 1 cm.

*Fibula*. The fibula is straight, long, and gracile (Figures 2 and 15). It is enlarged to form a head for fitting into the fibular fossa of the proximal part of tibia. The distal end is transversely flattened to accommodate the lateral tibia and the calcaneum.

*Pes*. The left astragalus and calcaneum are the only preserved tarsal elements. Generally, the astragalus is located directly below the tibia and medial to the calcaneum, and they are frequently fused together. In the holotype of *Minimocursor phunoiensis* gen. et sp. nov., the astragalus is bigger, but it is displaced below the calcaneum through taphonomy (Figures 2 and 15). From a possibly anterodorsal view, the astragalus is smooth, slightly concave in the middle of the plate, and quite inclined on the side to be attached to the calcaneum. The calcaneum is observed in the lateral view. It appears as a small nubbin of bone that articulates proximally with the fibula and tibia.

The possibly left metatarsals III and IV are present (Figures 2 and 15), but the proximal half of metatarsal IV is almost lost. They are long, slender and straight in lateral view as in *Agilisaurus*, *Hexinlusaurs*, and *Sanxiasaurus* [5,10,12] but different from those of *Lesothosaurus* and *Eocursor*, which are laterally curved distally [66,80]. The metatarsals seem to have an ovoid cross-section at the mid-shaft. The proximal and distal ends are expanded anteroposteriorly. Distally, the metatarsal III possesses a facet for accommodating the medial surface of metatarsal IV.

Although the pedal phalanges are missing, a single isolated ungual is preserved at the right lateral side of the specimen, above the right ilium (Figures 3 and 8). It is tapering, narrow, pointed, claw-like in lateral view, and dorsoventrally flattened.

### 3.2.5. Ontogenetic Assessment

Many distinct suture lines separate the neural arch from the centrum of the dorsal, sacral, and caudal vertebrae (Figure 8), indicating the holotype specimen of *Minimocursor phunoiensis* gen. et sp. nov. was an immature individual. Numerous isolated limb bones of various sizes, attributed to basal neornithischians were also found at the Phu Noi site, indicating that these dinosaurs were abundant in this area. We infer that the isolated bones of basal neornithischians from the Phu Noi belong to *Minimocursor phunoiensis* gen. et sp. nov. but of different ontogenetic stages. The femur length of the holotype is 8.2 cm (see Table 4). The body length of this dinosaur is estimated to have been about 0.6 m, comparable with *Agilisaurus* [12]. We assume that at an adult stage, it may have been up to two meters long, based on the longest femur found (about 26.6 cm in length).

**Table 4.** Selected measurements of *M. phunoiensis* gen. et sp. nov., based on the holotype PRC 150.

| Element | Measurement (mm) | |
|---|---|---|
| Cervical vertebrae (L) 8?, 9? | Length of ribs | 20.12, +23.10 |
| Dorsal vertebrae (L) 1–10 | Length of ribs | 49.85, 62.06, 66.54, +54.45, 65.53, 58.64, 45.98, 32.50, 17.83, 13.73 |
| Dorsal vertebrae (R) 12–15 | Length of centra | 10.54, 10.16, 10.76, 10.78 |
| Sacral vertebrae (R) 1–5 | Length of centra | 12.03, ?, ?, ?, 10.29 |
| Caudal vertebrae (L) 1–10 | Length of centra | 10.00, 9.98, 9.96, 9.74, 9.25, 9.25, 10.97, 9.02, 9.38, 10.22 |
| Scapula (L) | Length Proximal width Distal width Smallest diameter of shaft | +53.63 18.0 25.70 10.44 |
| Ilium (L) | Length Height Breadth of acetabulum | 70.37 14.66 7.26 |
| Pubis (L) | Length Length of preprocess Length of postprocess | 103.04 42.55 75.66 |
| Ischium (L) | Length Proximal width Distal width Smallest breadth shaft | 65.38 16.41 4.11 3.27 |
| Femur (L) | Length Proximal width Distal width Smallest shaft diameter | 82.08 14.07 19.69 9.78 |
| Tibia (R) | Length Proximal width Distal width Smallest shaft diameter | 99.76 14.72 19.84 7.64 |
| Fibula (L) | Length Proximal width Distal width Smallest shaft diameter | 95.41 9.64 7.38 3.74 |
| Metatarsal (L) ?III,?IV | Length | 53.83, 47.55 |
| Pes phalanx (?R) | Length | +20.55 |
| Pes ungual (?R) | Length | 11.12 |

### 3.2.6. Referred Material

The two following specimens, corresponding to ornithischians, were found in the same bed as the holotype PRC 150. They are tentatively referred to as *Minimocursor phunoiensis* gen. et sp. nov. (see phylogenetic analyses and discussion for comments).

Material: PRC 149 (renumbered from PN 13-09 in [20,21]), a lower jaw (Figure 16).

Comment: Buffetaut et al. [21] reported a lower jaw (PRC 149) from the Phu Noi locality. The total number of aveoli is not certain because the anterodorsal part is damaged, but at least twelve alveoli can be observed. There is a 'spout-shaped' decurve at the symphyseal area. The fan-shaped teeth have a strongly ridged crown and an asymmetric enamel distribution [21]. The dentary teeth of PRC 149 and other isolated teeth from Phu Noi are remarkable in having asymmetrically distributed enamel.

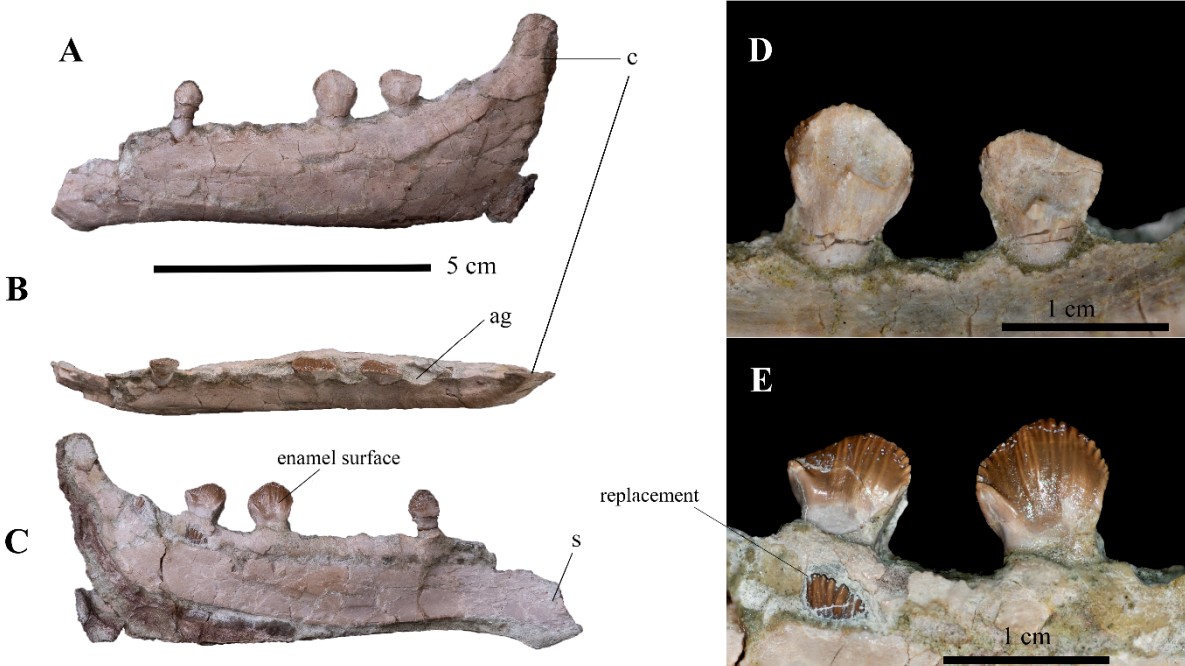

**Figure 16.** Left dentary (PRC 149) referred to *Minimocursor phunoiensis* gen. et sp. nov. from Phu Noi locality in labial (**A**), dorsal (**B**), and lingual views (**C**). Dentary teeth in labial (**D**) and lingual (**E**) views.

Material: SM2021-1-132, a left pes with tibia, astragalus, and calcaneum (Figure 17).

Comment: This left tibia is approximately 2.3 times longer than the tibia of the holotype. The tibia shaft is robust, long, straight, rounded in transverse section, and twisted along its long axis. Mt III is the longest, and Mt I is the shortest. The phalanges are long and slender. The claws are sharp and triangular, but not very recurved. Metatarsal V is not preserved. The left pes phalangeal formula is possibly 2-3-4-5-0?. The phalangeal formula is usually 2-3-4-5-0 in other neornithischians such as *Agilisaurus*, *Hexinlusaurus*, *Xiaosaurus*, *Hypsilophodon* and *Orodromeus* [9,10,12,60,74] (see Table 3)

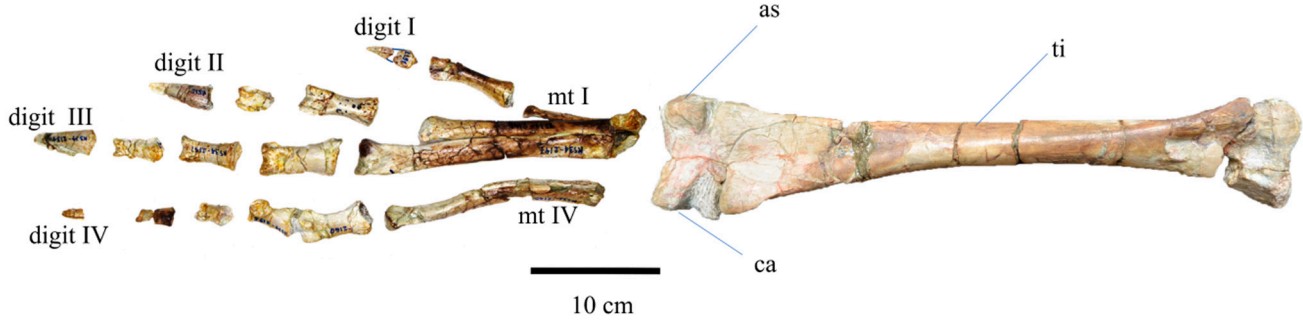

**Figure 17.** Left pes with tibia (SM2021-1-132) referred to *Minimocursor phunoiensis* gen. et sp. nov. from Phu Noi locality in dorsal view.

*3.3. Phylogenetic Analysis*

The first analysis (Figure 18), comprising all the taxa from the matrix of [39] and the OTU *Minimocursor*_gen.nov., was interrupted after the memory buffer was filled during the second round of RAS + TBR. It results in 50,000 MPTs (Most Parsimonious Trees) of 918 steps (RI = 0.638 and CI = 0.340). The strict consensus (1594 steps, RI = 0.263, CI = 0.202, Figure 18A) is largely unresolved. Several wildcard taxa are identified. Among them, *Laelly-nasaura* is either considered as a Heterodontosauridae, an Ornithopoda, or the sister group of *Yandusaurus* and *Hexinlusaurus*, resulting in the collapse of most of the nodes of the Ornithischia. A second wildcard taxon, *Notohypsilophodon*, is included inside Ornithopoda in most of the MPTs, but its position sometimes moves outside the clade including Cerapoda, Thescelosauridae, and Jeholosauridae, leading to major polytomies inside Neornithischia. These two genera are discarded in the second and third phylogenetic analyses.

The second analysis results in 21120 MPTs of 914 steps (RI = 0.641, CI = 0.341). As expected, the strict consensus (947 steps, RI = 0.621; CI = 0.329, Figure 18B) is much more resolved than that of the first analysis. The three OTUs corresponding to *Minimocursor* gen. nov. form a monophyletic group which is the sister group of a clade formed of Jeholosauridae, Thescelosauridae, *Nanosaurus*, *Hypsilophodon*, and Cerapoda. This result indicates that, even taken as independent OTUs, the holotype PRC 150 and the isolated lower jaw SM2021-1-132 are more closely related to each other than to any other OTU in the analysis, strongly suggesting that the holotype and the isolated lower jaw belong to the same taxon.

The third analysis also results in 21120 trees of 914 steps (RI = 0.641; CI = 0.341). The strict consensus (947 steps, RI = 0.621; CI = 0.329, Figure 19) is similar to that of the second analysis. The main ornithischian clades are retrieved, such as Genasauria, Thyreophora, Neornithischia, Cerapoda, Marginocephalia, and Ornithopoda. A few polytomies are still present, in Thescelosauridae (for *Thescelosaurus* spp. and most of the Orodrominae), non-iguanodontian Ornithopoda, Rhabdodontidae, and Dryosauridae. Note that even on the 50% majority-rule consensus (not figured here), most of these polytomies are not resolved. The Bremer supports are very low: except a very few nodes with a value of 3 or higher (i.e., Ornithischia and Herrerasauria), the Bremer supports are usually equal to 1 or, more rarely, 2.

*Minimocursor* gen. nov. shares several synapomorphies of Ornithischia, such as the dentary contributing to the coronoid process [80:1], subtriangular coronoid process [82:1], presence of a cingulum on dentary teeth [130:1], pubis posteroventrally rotated [194:2], 'pendant-shaped' fourth trochanter on the femur [219:2]. It shares with the Genasauria the 'spout-shaped' dentary symphysis [73:1] and the presence of fifteen dorsal vertebrae [147:1].

The main synapomorphies of Neornithischia (e.g., characters 10, 30, 31, 166, 200, 203) are scored as missing data in *Minimocursor* gen. nov., but it shares with *Agilisaurus* and its sister group several apomorphic features, including the well-developed coronoid process [79;1], the absence of an external mandibular fenestra [97:1], the pubic peduncle of the ilium tapered distally and smaller than the ischial peduncle [192:1], the rod-shaped prepubic process of the pubis [197:2], and the prepubic process of the pubis greater than 20% the total length of the ilium [198:1].

It shares with the clade formed of *Yandusaurus*, *Hexinlusaurus*, and their sister group the brevis shelf on the ilium extended medially in a roughly horizontal plane [189:1] and the lateral distal condyle of the femur representing 80–60% of the size of the medial distal condyle [232:1]. It shares with *Kulindadromeus* and its sister group the presence of a trench between the greater trochanter and the head of the femur [212:1].

*Minimorcursor* gen. nov. is the sister group of a clade formed of Jeholosauridae, Thescelosauridae, *Nanosaurus*, *Hypsilophodon*, and Cerapoda, based on the presence of denticles confluent with ridges extended to the base of the crown on dentary teeth [114:1], the presence of a lateral swelling of the ischiac peduncle of the ilium [191:1], and the dorsal margin of the lesser trochanter being approximately the same height as the head of the

femur [218:1]. It differs from all these taxa by the absence of a surangular foramen [85:0] (vs. present [85:1]) and a lesser number of vertebrae in the sacrum (five [148:1] vs. six or more [148:2/3]). It can also be excluded from the other main neornithischian clades such as the Marginocephalia due to the rod-shaped prepubic process of the pubis [197:2] (vs. dorsoventrally flattened [197:3]) and the Thescelosauridae due to the convex lateral surface of the greater trochanter of the femur [213:0] (vs. flattened lateral surface [213:1]).

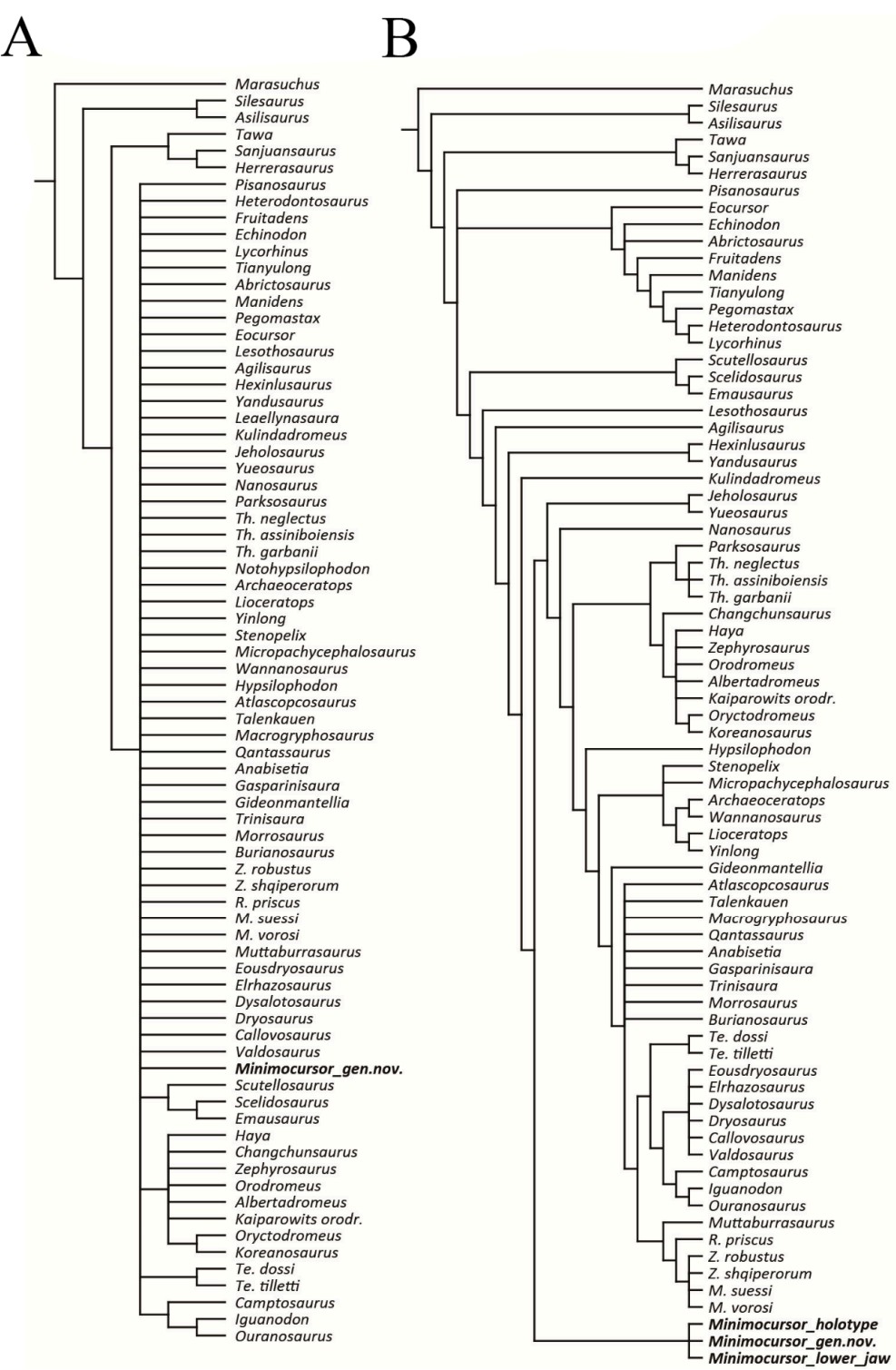

**Figure 18.** Strict consensus of 50,000 MPTs from the first analysis (1594 steps, RI = 0.263, CI = 0.202) (**A**). Strict consensus of 21,120 MPTS from the second analysis (947 steps, RI = 0.621; CI = 0.329) (**B**).

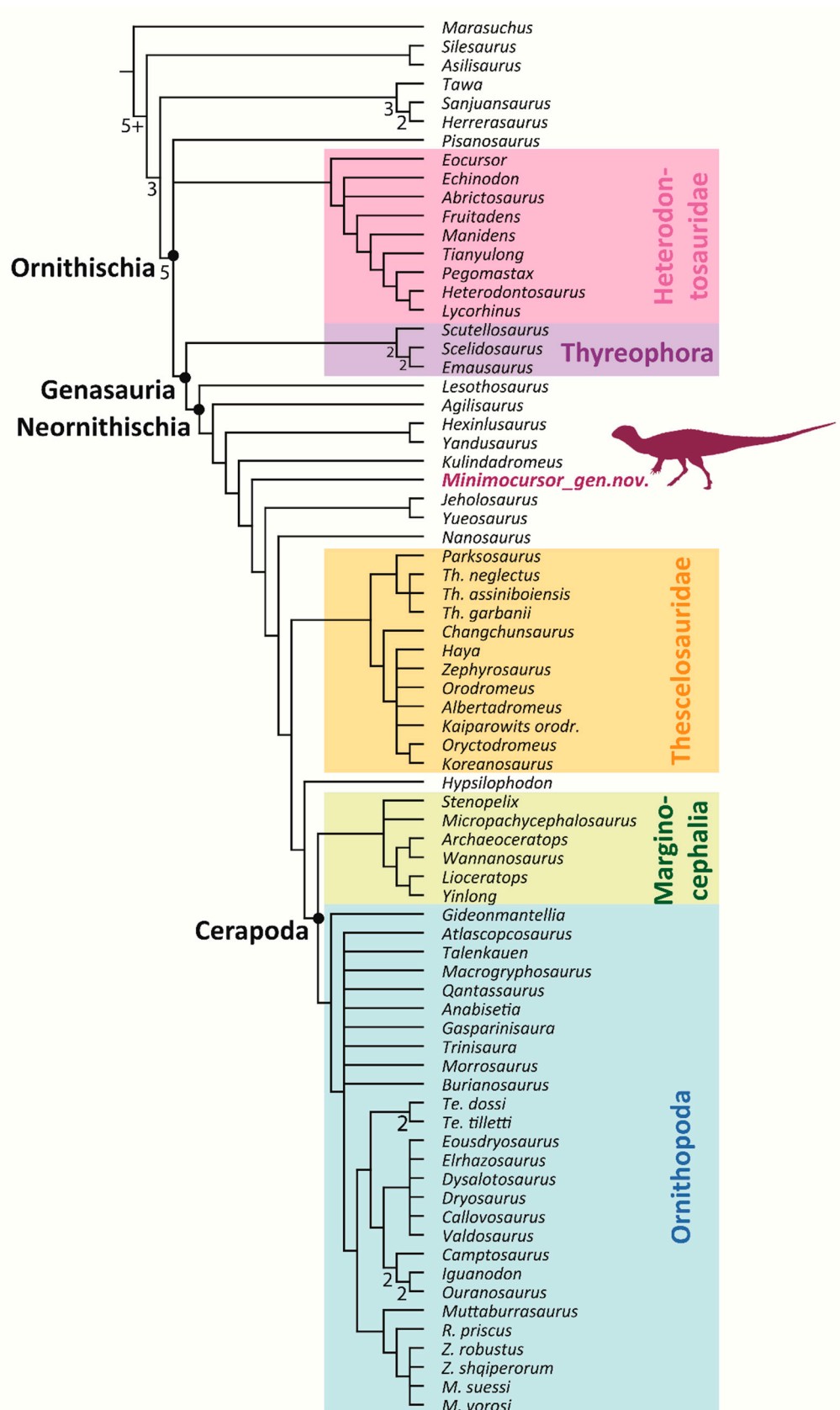

**Figure 19.** Strict consensus of 21,120 MPTS from the third analysis (947 steps, RI = 0.621; CI = 0.329). Node numbers indicate Bremer support greater than 1. Nodes without number have a Bremer support of 1.

## 4. Comparison and Discussion

*Minimocursor phunoiensis* gen. et sp. nov. shows a combination of both plesiomorphic and apomorphic characters of small basal neornithischians. One of the most distinct characters is the jugal boss, normally absent in other basal neornithischians and ornithopods [57]. In non-cerapodan neornithischians, it appears only in three thescelosaurids from the Cretaceous. This is the oldest evidence of this feature in basal neornithischians.

The dentary teeth of PRC 149 and other unpublished isolated teeth from Phu Noi are characteristic in having asymmetrically distributed enamel (see Table 5), as in a variety of more derived neornithischians [1,2,79,81], a condition different from *Agilisaurus* and *Hexinlusaurus*, which exhibit symmetrically distributed enamel [1,12]. The ornamented lingual side does not show a prominent median ridge contrasting with the European Early Cretaceous *Hypsilophodon* [81] and other derived neornithischians.

The pes phalangeal formula of *Minimocursor phunoiensis* is 2-3-4-5-0? as in other neornithischians such as *Agilisaurus*, *Hexinlusaurus*, *Xiaosaurus*, *Hypsilophodon*, and *Orodromeus* [9,10,12,60,74].

**Table 5.** Dental count and enamel distribution on tooth crowns of some ornithischians.

| Taxa | Premaxilla | Maxilla | Enamel Distribution | Mandible | Enamel Distribution |
|---|---|---|---|---|---|
| *Pisanosaurus mertii* | ? | 11+ (17–18?) | symmetry | 15 | symmetry |
| *Heterodontosaurus tucki* | 3 | 12 | asymmetry | ? | asymmetry |
| *Echinodon becklessii* | 3 | ~11 | symmetry | 10 | symmetry |
| *Tatisaurus oehleri* | ? | ? | symmetry | 18 | symmetry |
| *Lesothosaurus diagnosticus* | 6 | ~14 | symmetry | ~14 | symmetry |
| *Agilisaurus louderbacki* | 5 | 14 | symmetry | 20 | symmetry |
| *Hexinlusaurus multidens* | ? | 18 | symmetry | 20 | symmetry |
| *Xiaosaurus dashanpensis* | ? | ? | symmetry | ? | ? |
| *Yandusaurus hongheensis* | ? | 15 | asymmetry | ? | ? |
| ***Minimocursor phunoiensis*** | **?** | **?** | **?** | **≥12** | **asymmetry** |
| *Nanosaurus agilis* | ? | ? | symmetry | 13 | symmetry |
| *Hypsilophodon foxii* | 5 | 10–11 | asymmetry | 13–14 | asymmetry |
| *Convolosaurus marri* | 4 | 8–10 | asymmetry | 11 | asymmetry |
| *Dysalotosaurus lettowvorbecki* | 0 | 13 | asymmetry | 11–12 | asymmetry |

*Minimocursor phunoiensis* gen. et sp. nov. constitutes the oldest record of a neornithischian dinosaur in Southeast Asia so far. Numerous ornithischian remains formed of postcranial elements of various sizes have been found at Phu Noi and can be referred to this taxon, indicating that this dinosaur was abundant in this area. It lived together with mamenchisaurid sauropods, metriacanthosaurid theropods, and other vertebrates from the rich vertebrate assemblages of the lower part of the Phu Kradung Formation (Figure 20) [20,24,38,82]. The vertebrate remains support a Late Jurassic age for the lower Phu Kradung Formation [38,83,84], while the upper part of Phu Kradung Formation is Early Cretaceous in age based on the presence of *Dicheiropollis etruscus* [84]. Moreover, xinjiangchelyid turtles (such as *Phunoichelys kalasinensis* and *Kalasinemys prasarttongosothi*) and teleosaurid crocodylomorphs (such as *Indosinosuchus potamsiamensis*) from the lower Phu Kradung Formation are distinct from the assemblages of the upper part, which include abundant remains of more derived turtles such as trionychoids [30,31,85], pholidosaurids (such as *Chalawan thailandicus*), and atoposaurids [86,87], indicating a faunal turnover in Southeast Asia through the Jurassic–Cretaceous boundary [25].

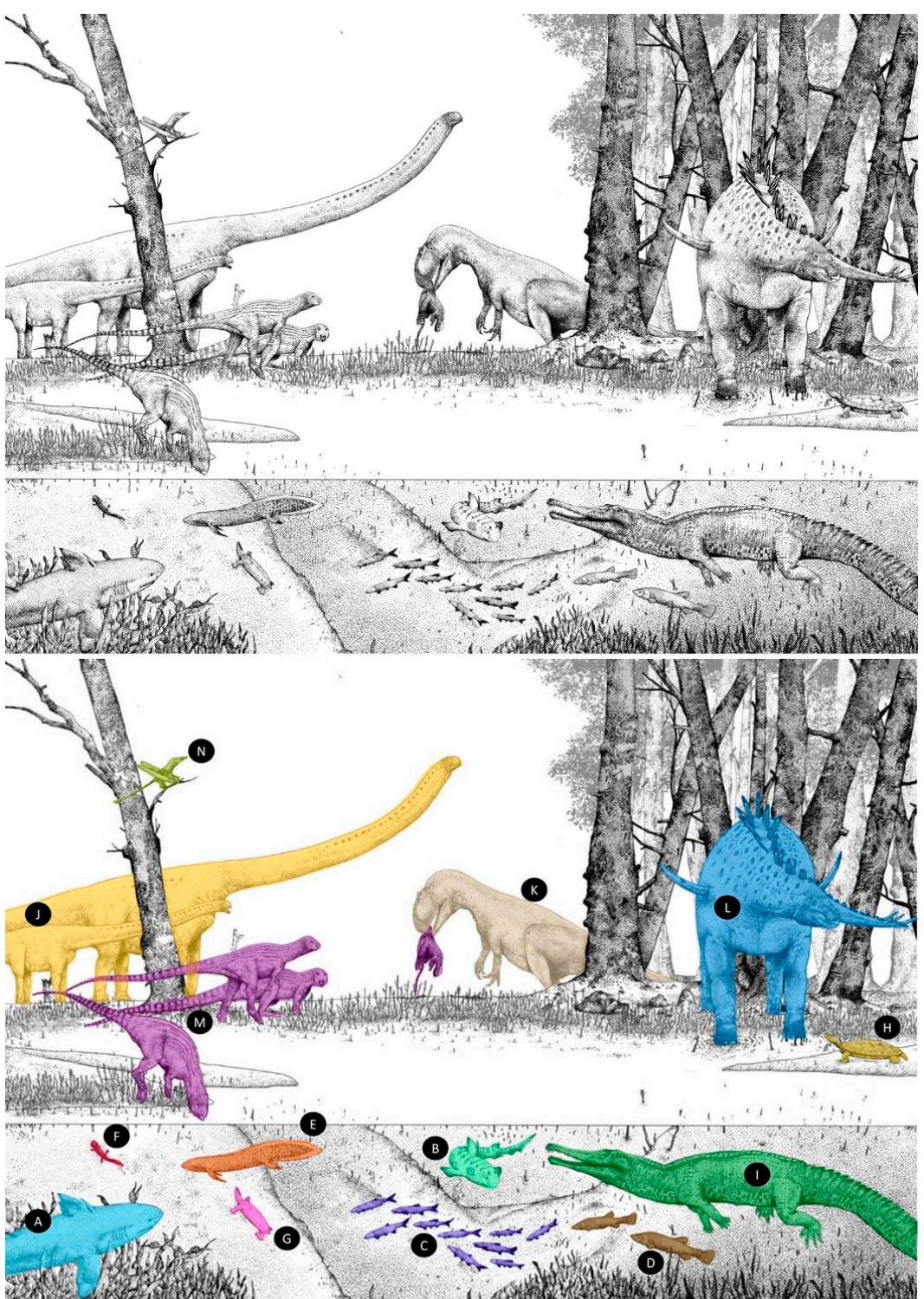

**Figure 20.** Palaeoenvironmental interpretation of the Late Jurassic Phu Kradung Formation of northeastern Thailand, including hybodont indet. with egg capsules (**A**), *Acrodus kalasinensis* (**B**), ptycholepid indet. (**C**), *Isanichthys lertboosi* (**D**), *Ferganoceratodus annekempae* (**E**), brachyopid indet. (**F**), *Phunoichelys thirakupti* (**G**), *Kalasinemys prasarttongosothi* (**H**), *Indosinosuchus potamosiamensis* (**I**), mamenchisaurid indet. (**J**), metriacanthosaurid indet. (**K**), stegosaur indet. (**L**), *Minimocursor phunoiensis* (**M**), and rhamphorhynchoid indet. (**N**). Drawing by Sakka Weerataweemat.

## 5. Conclusions

The Phu Noi locality contains a wealth of specimens and has yielded an exceptionally articulated skeleton, which represents one of the best-preserved dinosaurs ever found in Southeast Asia. This is the earliest record of neornithischians in Southeast Asia, and the first dinosaur taxon named from the Phu Kradung Formation of Thailand. This finding increases diversity and helps to elucidate the evolution of basal neornithischian dinosaurs in this region. Many of the remaining bones are still under preparation, including another skull. These unpublished specimens may provide a better understanding of the biology of *Minimocursor phunoiensis* gen. et sp. nov. in the future.

This study also provides new palaeontological data to illustrate the palaeoecosystem to the general public, as well as improving the academic value of the Kalasin Geopark.

**Author Contributions:** Conceptualization, S.M., U.D., P.C. and E.B.; methodology, S.M., U.D., P.C., V.S. and W.B.; software, S.M. and B.K.; validation, S.M., U.D. and P.C.; formal analysis, S.M., B.K. and T.N.; investigation, S.M., U.D., P.C., W.B. and V.S.; resources, S.M., U.D., V.S., P.C. and W.B.; data curation, S.M., P.C. and W.B.; writing—original draft preparation, S.M.; writing—review and editing, S.M., E.B., U.D., P.C., V.S., T.N., B.K. and W.B.; visualization, S.M., E.B., U.D., P.C., V.S. and W.B.; supervision, S.M. and E.B.; project administration, S.M., U.D. and P.C.; funding acquisition, S.M. All authors have read and agreed to the published version of the manuscript.

**Funding:** Thailand Science Research and Innovation (TSRI): FF-660632/2566. International Research Network "Paleobiodiversity in South-east Asia" of the Centre National de la Recherche Scientifique (Paris).

**Institutional Review Board Statement:** Not applicable.

**Data Availability Statement:** Not applicable.

**Acknowledgments:** We are grateful to all the staff of the Palaeontological Research and Education Centre of Mahasarakham University, and Sirindhorn Museum, Department of Mineral Resources, who took part in fieldwork and helped during visits to the collections. Many thanks to You Hai-Lu and all the staff of the Institute of Vertebrate Palaeontology and Palaeoanthropology, Chinese Academy of Sciences (IVPP) for allowing access to specimens in Beijing. Assistance with the map was provided by Chanon Suriyonghanphong, and Kantanat Trakunweerayut. The skeletal reconstruction is from Wongwech Chowchuvech. The Phu Noi ecosystem is illustrated by Sakka Weerataweemat. The TNT software for phylogenetic analysis is made freely available by the Willi Hennig Society. We would like to thank three anonymous reviews for their helpful comments on an earlier version of the manuscript. This research project was financially supported by Thailand Science Research and Innovation (TSRI). Sita Manitkoon was financially supported by the Mahasarakham University Development Fund for the international symposium presentation. Bouziane Khalloufi is grateful to the Foundation Ars Cuttolli, Paul Appell, for its support. Eric Buffetaut's participation was supported by the International Research Network "Paleobiodiversity in South-east Asia" of the Centre National de la Recherche Scientifique (Paris).

**Conflicts of Interest:** The authors declare no conflict of interest.

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
