# Peer review of "A New Basal Neornithischian Dinosaur from the Phu Kradung Formation (Upper Jurassic) of Northeastern Thailand"

_diversity, doi:10.3390/d15070851_

Round 1

Reviewer 1 Report

Manitkoon et al. A new basal neornithischian from northeastern Thailand

The authors present an articulated postcranial skeleton of a small, perhaps juvenile, ornithischian dinosaur from northeastern Thailand. Several such specimens are known from other Asian localities, most notably the Lower and Upper Shaximiao Formations of the Sichuan basin, PR China. However, this specimen represents the best-preserved basal ornithischian from SE Asia (an area historically undersampled for dinosaur fossils). Given its geographic location, the specimen probably represents a new taxon. Unfortunately, the paper is not detailed enough to determine whether the specimen is should be named as a new taxon or not.

The authors claim the taxon can be diagnosed by a unique combination of characters, but they do not discuss why this combination is unique. As far as I can tell, all of the characters they mention in the diagnosis are common to a range of basal ornithischians and non-cerapodan ornithischians, and are probably symplesiomorphies. A section is needed under the diagnosis discussing the distribution of these characters in other taxa, to demonstrate that they are not found in any other combination. The autapomorphy that they authors claim to have identified also appears to be present in Lesothosaurus (see Baron et al. 2016) and thus is also likely to be a symplesiomorphy. The diagnosis cannot, therefore, justify a new taxon as it stands.

The description is extremely brief and the paper is poorly illustrated. The descriptive information doesn’t really help us to understand the anatomy in a lot of detail. There are many places in the description where words like ‘short’ or ‘small’ are used, but these are relative terms. When describing shapes, e.g., “the spine is triangular in shape”, specify the orientations and in what view: the spine is triangular in shape in lateral view, with the apex pointing dorsally. Instead of using terms such as ‘higher’ and ‘lower’, use anatomical orientations: located more dorsally on the neural arch; located ventral to… etc. High-resolution photographs of all the elements are needed, and an interpretive drawing of the pelvis would also be useful. I’ve made some comments on the description below, but I have been unable to identify any feature that seems unique about this skeleton from it. I’m afraid, based on this description, the specimen is indeterminate and should not be named.

The phylogenetic analysis section warrants more detail. Section 3.3 should be in the methods. There are several competing phylogenies of basal ornithischians out there, of which Baron et al. (2017) isn’t even in the top four. So I’d like to know why the authors chose this one, instead of Han et al., Boyd et al., Dieudonne et al., or Butler et al.? Some discussion about the choice of phylogeny is warranted, or the taxon should be put in more than one phylogeny. The authors present a majority rule tree. Majority rule trees are unsupportable. The only tree that should be presented is a strict consensus tree, and support measures, preferably Bremer support metrics, should be presented.

Minor comments

Line 64: “excavated remains from the solid rock…” would be better phrased as well-consolidated rock?

Line 82-85: .. it is characterized by brownish-purple and greenish-grey sandy siltstones which were deposited on a natural levee with low energy. The elevation of the bone bed is about 243-258 meters”. Is the elevation above sea level? If so, say this.

Line 86: THE Phu Noi locality…

Line 95: crocodylomorphs

Line 103: The vertebrate fauna resembles that of the Middle to Late Jurassic…

Line 128: neornithischians is incorrectly spelled

Line 132: first word “tata” should be “taxa”

Line 135: Euparkeria capensis should be italicized

Line 175: Ornithischia is misspelled

Diagnosis: Why mention the ossified tendons? They’re quite a common feature in basal ornithischians and ornithopods.  

Line 197: What are “restricted” ossified tendons? In what way are they restricted?

Caption for Fig. 5: “Recovered elements of the holotype”

Table 1: There’s quite a lot of debate about the age of the Lower and Upper Shaximiao Formations, and it seems clear that they are time-transgressive across their outcrop areas. Wang et al. (2018) dated the Lower Shaximiao to Oxfordian, while Dai et al. (2021) dated it to younger than Bajocian. I think it’s reasonable therefore to describe the ages of Agilisaurus, Hexinlusaurus, and Xiaosaurus as Middle to Late Jurassic and Yandusaurus as Late Jurassic. Given the Oxfordian age from Wang et al., the Upper Shaximiao could easily be Kimmeridgian or Tithonian. I don’t think we have a good handle on that and I don’t think you should quote the ages as being more accurate than they are.

In contrast, the exposed Wessex Formation is Barremian in age, so you should remove the Berriasian– part of that – it’s clearly Barremian.

Description

Line 228: Why is tooth listed under the axial skeleton? It should be described before that heading. Neither the description of the tooth nor the figure is sufficient. The figure is small and blurry – please provide a high-resolution close up figure of the tooth. I don’t know what the “fan-like shape” means. Are there ridges? Denticles? Is there a wear facet? These features are phylogenetically important and need to be described in detail.

Line 232: Why is there a number 1 in the middle of the first sentence? Please provide a higher-resolution figure of the cervical vertebrae. The details can’t be seen on Fig. 2. If the vertebrae are articulating, how can you see their posterior ends? Surely they’d be visible in lateral view?

Line 234: On what basis are you suggesting the specimen had nine cervical vertebrae? These vertebrae aren’t preserved, so this is unwarranted speculation.

Line 235: Agilisaurus should be italicized

Line 238: Mentioning the opisthocoelous condition in other taxa is pointless because you can’t see the ends of the centra in your taxon. Remove this.

Line 241: appears to repeat the same information as line 237.

Line 243: The diapophysis is the articular facet for the rib. It is supported on a transverse process that in basal ornithopods usually projects horizontally. The statement “The diapophysis is approximately 90 degrees from horizontal” therefore makes no sense in terms of the anatomy of dinosaurs.

Line 244: The position of the diapophyses seems higher up the neural arch than what?

Line 252: Why are only dorsals 12 to 15 described in lateral view? The abbreviation for dorsals is variably given as dc and dv. Choose one and be consistent.

Line 262: Are the neurocentral sutures actually unfused, or is it just that you can see the suture still? If the latter, it doesn’t necessarily indicate the animal is a juvenile, just that it wasn’t necessarily fully osteologically mature, but to determine this properly you’d need to do histology.

Dorsal vertebrae:  Can you describe more details of the vertebrae? What about the postzygapophyses and prezygapophyses? Can you provide a higher-resolution figure of the vertebrae?

Line 275: Don’t speculate on the shape of something if you can’t see it. It would be useful to provide a table of measurements of all elements of the skeleton, including the lengths of all vertebral centra, rather than just vague statements about size getting smaller.

Line 279: These are usually referred to as transverse processes or caudal ribs, not diapophyses as the diapophysis is the rib facet, not the process itself.

Scapula: It would help to describe the orientation in which the scapula is being described. I think you are describing it with the long axis vertical, but it would be helpful to specify that as it varies from description to description. It looks like the proximal plate of the scapula is not preserved. Please provide a higher-resolution photograph so that can be seen. If the proximal plate is preserved, please describe it. How can you determine there is damage to the coracoid facet (generally called the glenoid) if it’s not preserved?

Manus: what shape are the radiale and intermedium? Saying they are present isn’t a description. The metacarpals of digits III and IV are the longest in Lesothosaurus (see Baron et al. 2016), so it’s possible this is actually a plesiomorphy rather than an autapomorphy for this dinosaur.

Line 356: This isn’t a suture, it’s a breakage. The preacetabular process and postacetabular process aren’t separate bones. An interpretive drawing would greatly help understanding of the pelvic region. It’s difficult for me to see the supra-acetabular flange.

Line 363: All basal ornithischians, and indeed many ornithodirans, were bipedal. I don’t know what the evidence for the supraacetabular flange being lost helping with bipedalism is, but it sounds like rubbish to me.

Hind limb: please provide high resolution photographs of the hind limb elements. With regard to the pes, please don’t describe features in other dinosaurs that aren’t present in your taxon as it doesn’t add anything to the description.

Line 472: Upper Shaximiao

Figure 10: It would be useful to show a close up high resolution image of the teeth.

Section 3.3 phylogenetic analysis: Much of this should be in the methods.

Line 533: You refer to the strict consensus tree and cite Fig. 12, but according to the figure caption, this is a majority rule tree, not a strict consensus. Majority Rule trees are basically pointless. You should present a strict consensus tree.

Line 552: The pendant fourth trochanter is not unique to basal neornithischians. It’s present in virtually all bipedal ornithischians, including basal thyreophorans and heterodontosaurids, both of which lie outside of Neornithischia.

The English is okay. I've picked up a few spelling mistakes and suggested a few re-wordings but haven't exhaustively copy-edited it. I think the remaining mistakes can be picked up by a read through and a copy-editor. 

Reviewer 2 Report

The authors describe a new neornithischian dinosaur from the Upper Jurassic of Thailand based on a well-preserved specimen. I agree to establish a new genus and species, although the skull is not preserved and no autapomorphy (the unique feature of Minimocursor is actually not correct, see below). Most parts of the description and pictures are clear, but there are also some mistakes, I think this manuscript can be improved to be more accurate and readable. Here are my comments listed as follows:

1. The new specimen needs to compare to more small ornithischians, especially the complete specimens, such as Jeholosaurus (Han et al 2012), Haya (Barta and Norell, 2021), Changmiania (Yang et al 2019), and Changchunsaurus (Butler et al. 2011)

2. For phylogenetic analysis, I suggest using a more recent data matrix of ornithischians, such as Barta and Norell 2021 (DOI: 10.1206/0003-0090.445.1.1) or Sues et al. 2022 (.DOI: 10.1016/j.cretres.2022.105369). In addition, I suggest providing a reduced strict consensus tree, which is more reliable than 50% majority rule tree.

3. The description of the manus and Figure 9 have some mistakes, especially the labelled metacarpal are all wrong based on my observation of the photographs and the comparison with Hexinlusaurus. The labelled “mc V” and “mcIV” should be “mc I” and “mcII”, respectively. Again, The labelled “mc II” and “mc I” should be “mc IV” and “mc V”. Therefore, the author's description that mc III and IV are the longest is not correct and also not a diagnostic feature any more. In fact, mc II and III are the longest as in most other ornithischians.

4. Please also provide the measurements of vertebrae if possible.

5. In Figure 2, line drawing of the scapula seems not consistent with the photograph. Weirdly, there is a rodlike process extending posteroventrally at the preserved proximal region, and also the distal end extends more posteriorly than that in the photograph.

6. In Figure 3, I suggest removing the rock matrix that does not have bones and zooming in the skeleton, Moreover, please label and cite importable features in the photograph.

7. In Figure 4, the isolated tooth is not clear in its morphology and should be retaken.

8. P7, line 201, the diagnosis mentioned some features in the pelvic girdle, which should be marked in Figures 2 and 6.

9 In Table 1, as you have already added Hypsilophodon, please also add Jeholosaurus, Changmiania, and Changchunsaurus (all from the Early Cretaceous of China)

10. In Figure 6, please add a scale, and label important features in pelvic girdle and sacral vertebrae.

11. The description of vertebral column is a little simple. I suggest adding more comparisons with heterodontosaurids and other small ornithischians such as Hypsilophodon, Jeholosaurus, Haya, Changchunsaurus, Parksosaurus (Sues et al. 2023).

12. P12, line 315, what is the “ventral protuberance of the scapula”? Please label it in the figure. Is this feature special? Please also add more comparisons with other small ornithischians.

I have a few minor comments in the attached pdf.

I think its fine.

Reviewer 3 Report

Sita Manitkoon and colleagues present and gorgeous new ornithischian from Thailand. The holotype is well-preserved and an important addition for the evolutionary tree of Ornisthichia. I agree with the main finds of the study. Moreover, the manuscript is well-organized, clear and concise. In my opinion, after some minor modifications, the manuscript will be ready for publication. I marked some minor suggestions in a PDF. My main concerns/suggestions are:

            • In the “Material and methods” item you should include all the procedures employing in the phylogenetic analysis (software, version, parameters). Some of these points are listed below. Please, move to “Material and methods”. Moreover, regarding the phylogenetic analysis, I would like to see more information. Please, expands the results, describes the position of the new taxon in the strict consensus tree (and insert a figure depicting the strict consensus tree instead only the 50% majority rule tree). Some support values are welcome too. For instance, the Bremer support values are almost mandatory. Bootstrap indexes are welcome too. Some comments regarding the synapomorphies that support the position of the new species?

            • The holotype is a skeletally immature individual. So, please, insert an “Ontogenetic Assessment” sub-item in the “Systematic Paleontology” item. In “Ontogenetic Assessment” you will indicate the presence of visible neurocentral suture in the holotype as an indicative of a non-skeletally immature individual.

• The holotype is an amazing specimen. I would like to see more detailed images of the structures. For instance, a detailed image of the pelvic girdle, indicating the structures. A detailed image of the femur and femoral head depicting the structures.

Congratulations, the study is nice and offers a pleasant reading.

All the best,

Round 2

Reviewer 1 Report

The authors are to be congratulated on the substantial improvement to the description, which is now far more detailed, and on the figures, which are now really useful and clear. I have a couple of minor comments: 

Lines 53 and 54: Capitalization of Lower Shaximiao and Upper Shaximiao is needed as these are formal formation names. I refer to my earlier review in which I suggested that you should indicate that at least the Upper Shaximiao Formation is probably Late Jurassic in age (Wang et al. 2018), so you should change line 42 to read “Middle to Late Jurassic of Sichian, China”

Line 57: replace “advanced” with “later diverging”

Line 65: “from an older formation”

Line 132: “an articulated specimen that is more than..”

Phylogenetic analysis: 100 replicates is almost certainly not enough in a heuristic search to fully search tree space. Have you tried more? It might not make a difference, but you should try it. It might be that there are more parsimonious solutions that are out there in the tree landscape. You could also try a new tech search, followed by a round of TBR branch swapping with trees held in memory, as the new tech search algorithms search treespace more fully and are more likely to recover the most parsimonious topologies.

Diagnosis: when you say “apomorphic”, presumably you mean “autapomorphic”? please replace this. Can you indicate which characters are autapomorphies? This is often done using an asterisk.

Line 742: “This supraacetabular flange…” Move the sentence starting thus, which you have added in, to the section on the supraacetabular flange, before the description of the brevis shelf.

Line 770 (and also in the diagnosis): in figure 13, the prepubis really doesn’t look like it’s extending further anteriorly than the preacetabular process of the ilium. I’d say the preacetabular process extends further anteriorly.

Line 900: Referred material. Dentary - You can’t really refer this, as it doesn’t overlap with the holotype. I’d suggest you are a bit more tentative here, and say that it’s possibly referrable to the same taxon.

Line 1044: repetition of rod-shaped – you say this twice in the sentence. 

Improvements to the English could certainly be made but I think these could be done by the handling editor and copy-editors. 

Reviewer 2 Report

Thanks for your new work. The modified manuscript  was improved a lot and is suitable for publication. Nevertheless, I find some minor things that need to clarify or modify.

Is the posterior ramus of jugal bifurcated? it seems to be true based on photograph in Fig. 5, but the reconstruction is different. Please explain it. Moreover, there is no explanation of "rap" in Anatomical Terminologies.

In Fig. 9, what does the word "Ir (lateral ridge) and hg(haemal groove) mean? Please provide description and  expalanation in the text.

In Fig. 13, There is no explanation of "por" in Anatomical Terminologies.

In Fig. 15, Picture "A" should be in posterior view.

In addition, I also suggest to provide a "nex" or "tnt" file of data matrix as supplementary files if possible. This will be convenient for other people to reivew and use. 
